**Subject Category:**
Biology (whole organism)

ecology

carnivores, coexistence, depredation, interviews, occupancy modelling, sign surveys

**Author for correspondence:**
Arjun Srivathsa
e-mail: asrivathsa@wcsindia.org

# Examining human–carnivore interactions using a socio-ecological framework: sympatric wild canids in India as a case study

Arjun Srivathsa[1,2,3,4], Mahi Puri[2,3,4], Krithi
K. Karanth[4,5,6], Imran Patel[4] and N. Samba Kumar[3,4]

[1]School of Natural Resources and Environment, and [2]Department of Wildlife Ecology and Conservation, University of Florida, Gainesville, FL, USA
[3]Wildlife Conservation Society – India, Bengaluru, India
[4]Centre for Wildlife Studies, Bengaluru, India
[5]Wildlife Conservation Society, New York, NY, USA
[6]Environmental Science and Policy, Nicholas School of the Environment, Duke University, Durham, NC, USA

AS, 0000-0003-2935-3857

Many carnivores inhabit human-dominated landscapes outside protected reserves. Spatially explicit assessments of carnivore distributions and livestock depredation patterns in human-use landscapes are crucial for minimizing negative interactions and fostering coexistence between people and predators. India harbours 23% of the world's carnivore species that share space with 1.3 billion people in approximately 2.3% of the global land area. We examined carnivore distributions and human–carnivore interactions in a multi-use forest landscape in central India. We focused on five sympatric carnivore species: Indian grey wolf *Canis lupus pallipes*, dhole *Cuon alpinus*, Indian jackal *Canis aureus indicus*, Indian fox *Vulpes bengalensis* and striped hyena *Hyaena hyaena*. Carnivore occupancy ranged from 12% for dholes to 86% for jackals, mostly influenced by forests, open scrublands and terrain ruggedness. Livestock/poultry depredation probability in the landscape ranged from 21% for dholes to greater than 95% for jackals, influenced by land cover and livestock- or poultry-holding. The five species also showed high spatial overlap with free-ranging dogs, suggesting potential competitive interactions and disease risks, with consequences for human health and safety. Our study provides insights on factors that facilitate and impede co-occurrence between people and predators. Spatial prioritization of carnivore-rich areas and conflict-prone locations could facilitate human–carnivore coexistence in shared habitats. Our framework is ideally suited

for making socio-ecological assessments of human–carnivore interactions in other multi-use landscapes and regions, worldwide.

# 1. Introduction

Global carnivore distributions overlap highly with human-use landscapes [1]. Multi-use heterogeneous landscapes can serve as important subsidiary habitats for supporting populations of several carnivore species, and therefore offer great conservation potential [2]. Unfortunately, most current conservation strategies are focused on protected reserve creation and management, particularly in developing countries [3,4]. This is problematic because protected reserves in a majority of countries cover only 4–11% of the land area [5]. Furthermore, the socio-cultural, financial and political challenges that plague management of large carnivores in human-use areas makes it difficult to formulate policies that ensure their conservation while also safeguarding human lives, property, livelihoods and well-being [6].

Spatial overlap between carnivore distributions and human-use areas increases human–predator interface, resulting in negative interactions. Such scenarios are more common in developing countries where people's livelihoods are directly dependent on land and livestock [7]. People depend on forests for wood and other non-timber products [8,9]. Multi-use forests may also serve as grazing lands for domestic livestock [10]. On the other hand, carnivores foray into farmlands, villages and, sometimes, even large cities [11], thereby creating contentious 'shared spaces' between people and predators. As a result, humans face livestock losses, threats to life, and missed opportunity costs from avoiding areas with carnivore presence. Carnivores, in turn, face injury, retaliatory killing or physical removal following livestock depredations or human attacks. In many cases, people's negative attitude towards carnivores is also from perceived threat rather than actual losses [12,13].

With the global increase in human population and consequent impacts on wildlife, anthropogenic activities can potentially facilitate or impede carnivore persistence in shared spaces [7]. India harbours around 23% of the world's carnivore species in approximately 2.3% of the global land area. These carnivores share space with a population of 1.3 billion people, where human densities are 400 people $km^{-2}$ on average. Protected reserves constitute about 4% of the country, and roughly 19% of the land area has unprotected forest cover. Such forests, together with a multitude of non-forest habitats (agroforests, scrublands, barrenlands, grasslands, etc.), harbour populations of large carnivores outside the reserve network [14]. Wild canids, in particular, are a case in point. Although India's five canid species and three subspecies show widespread distribution across diverse landscapes [15], few studies have undertaken ecological assessments or evaluated their conservation requirements in shared habitats dominated by human activities. Given the range of risks they face, and the potential for conflict between wild canids and humans, such assessments could benefit both people and predators.

We examined factors that facilitate coexistence between humans and four sympatric wild canid species (Indian grey wolf *Canis lupus pallipes*, dhole *Cuon alpinus*, Indian jackal *Canis aureus indicus*, Indian fox *Vulpes bengalensis*) in the Kanha–Pench forest landscape of central India. We also included the striped hyena *Hyaena hyaena* in our assessment because they have somewhat similar ecological requirements. In a recent study, Gálvez *et al.* [16] propose a framework combining ecological and social information for examining human–carnivore interactions and identifying conservation measures. We adapted and expanded this framework, formally integrating ecological and social dimensions of carnivore conservation in human-dominated landscapes (figure 1). Based on our findings, we identified areas that would warrant spatial prioritization, and provide management recommendations for reducing wildlife-related losses to people while also conserving the carnivore community in this landscape.

# 2. Material and methods

## 2.1. Study framework

The framework we use follows four sequential steps (figure 1). In the first step, we identified ecological attributes that would influence distribution of the five carnivore species. We built a candidate set of models for each species based on *a priori* predictions. Analysis of field-based sign survey data generated spatial probabilities of presence (distribution) for the five species. The second step relates to the social dimension. Data on carnivore presence, depredation events and socio-economic attributes of

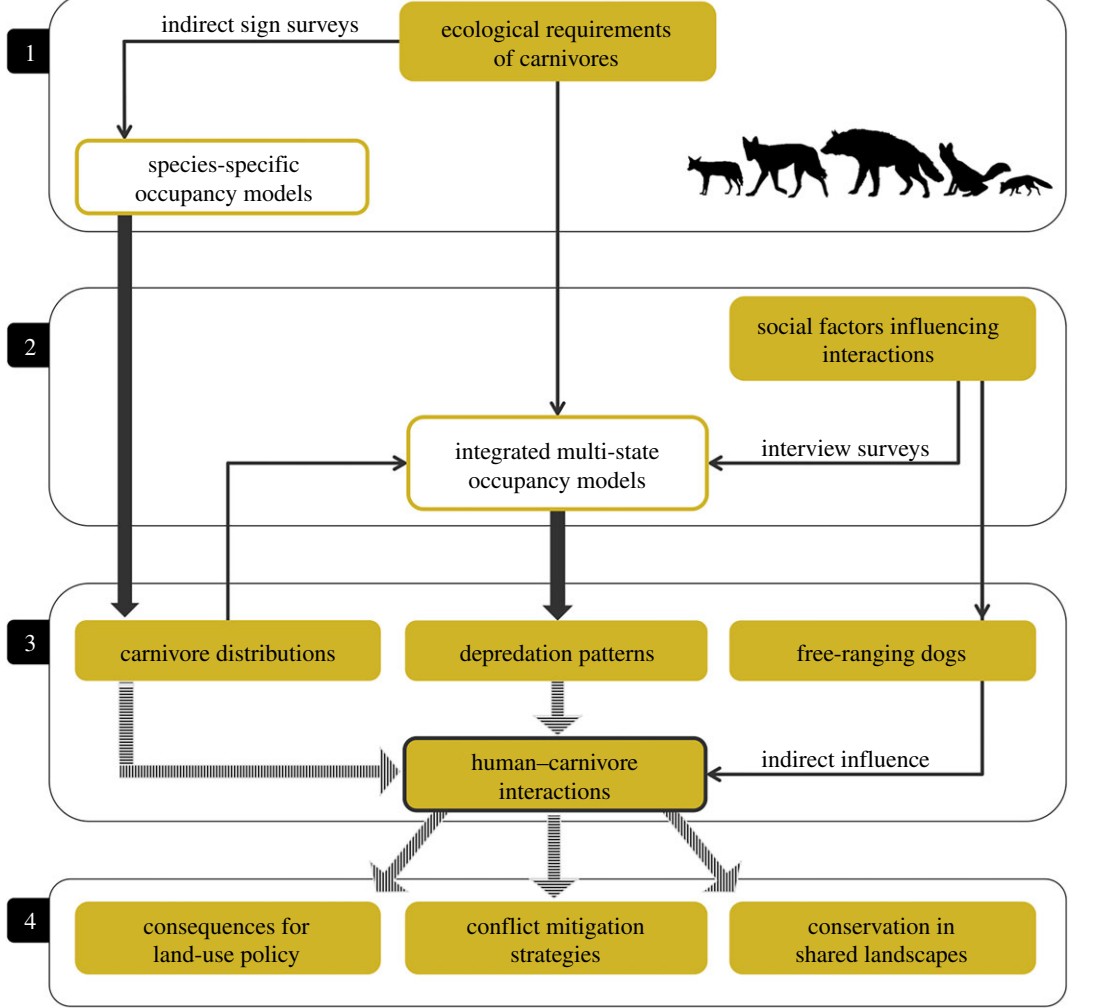

**Figure 1.** An integrated socio-ecological framework to examine human–carnivore interactions in shared landscapes. The four panels represent sequential steps, clear boxes contain statistical modelling approaches, line arrows are processes or contributing factors, block arrows indicate model outputs, and shaded arrows show inferences or implications.

people in the landscape were collected through questionnaire-based interview surveys of local residents (at the same spatial scale as the sign surveys). We tested the influence of ecological variables, social factors and species distribution (from the previous step) on depredation patterns. Step 3 represents the outcomes from the first two steps and also allows for incorporating ancillary information that would together contribute towards understanding human–carnivore interactions. In our case, the ancillary data pertain to distribution of free-ranging dogs. This step could also incorporate other factors that are not directly included in the modelling process (e.g. mortality, harvest, tolerance levels, etc. as relevant to the context). The final step (4) contains plausible inferences that could be drawn from all aspects that contribute towards human–carnivore interactions. This may include management implications, policy recommendations, spatial prioritization or refinement of methods/models used in steps 1–3.

## 2.2. Study area

The Kanha–Pench forest landscape (22°17′31.1″ N, 79°59′49.5″ E) extends over *ca* 160 km between Kanha (940 km²) and Pench (411 km²) Tiger Reserves in the southern part of the State of Madhya Pradesh, India (figure 2). The approximately 10 000 km² landscape harbours dry deciduous forests interspersed with grasslands, scrublands and agricultural lands. These habitats support populations of the five focal carnivore species (hereafter referred to as wolf, dhole, jackal, fox and hyena). The region also harbours other carnivores like the tiger *Panthera tigris*, leopard *P. pardus* and sloth bear *Melursus ursinus*, along with a suite of large ungulate herbivores. The landscape has a large number of human habitations,

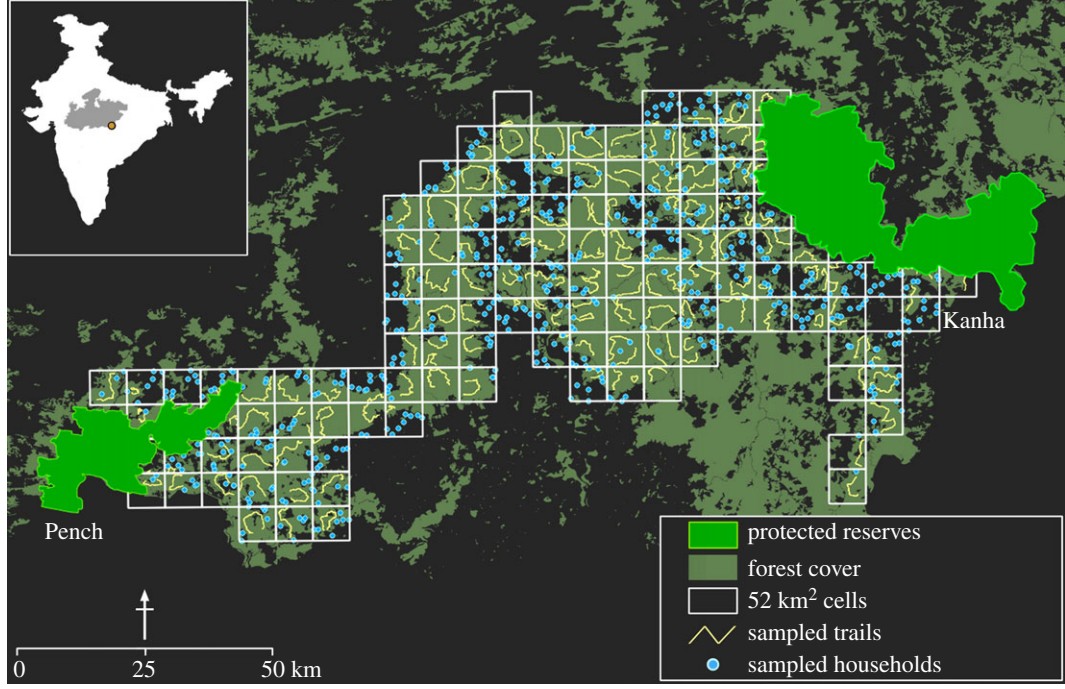

**Figure 2.** Study area and design for sign surveys and questionnaire-based interview surveys to examine human–carnivore interactions in the Kanha–Pench forest landscape, 2015–2016. The map shows forest cover, protected reserves, grid network with 128 cells of 52 km² size each, and surveyed forest trails and households. Inset: location of the study area in the State of Madhya Pradesh (grey), India.

with over 400 villages including several ethnic tribes inhabiting the forest interiors. Agriculture is the cornerstone of rural economy, but collection of non-timber forest resources, small-scale mining, and wage labour in nearby towns supplement household incomes.

## 2.3. Survey design

We overlaid a grid network with 128 cells of 52 km² each across the landscape (figure 2). Each cell was treated as an independent sampling unit. Cell size was chosen based on ecological, logistical and sampling considerations, that included home range sizes of the study species, spatial coverage of the study area and adequate sample sizes. We used an occupancy-based framework to assess distribution and depredation probabilities of the five species [13,17]. The occupancy metric is sensitive to the spatial scale at which assessments are made [18,19]. Based on the size of cells (henceforth 'sites') relative to the home range sizes of the five species, we interpret the occupancy parameter ($\psi$) as 'habitat-use probability' for wolf and dhole (home range sizes of the two species are greater than 52 km²; see [20] for wolf; [21] for dhole). For jackal, fox and hyena, we interpret $\psi$ as 'true occupancy probability' (home range sizes of the three species are less than 52 km²; see [22] for jackal; [23] for fox; [24] for hyena).

## 2.4. Field surveys

We surveyed forest roads and trails for indirect signs of the five carnivores from October 2015 to January 2016. Three survey teams (trained in field identification of large carnivore signs) searched for scats and tracks of all the focal species, following field protocols described in [17]. The teams used reference photographs, measurements and secondary/ancillary signs to ascertain species identity. Direct sightings of species were excluded from the analyses to retain uniformity in the detection process. Detection/non-detection (1/0) data were recorded for each species along contiguous 1 km trail segments; each segment was treated as a spatial replicate. Walk effort ranged from 2 to 23 km per site, proportional to the forest cover in the corresponding site. The five species vary in body size (wolf—25 kg; dhole—15 kg; jackal—10 kg; fox—3.5 kg; hyena—36 kg) and occur in a range of habitat types [15]. Surveying along forest roads/trails allowed us to maximize the probability of detecting their signs. Along with data related to the five carnivore species, we also recorded signs of ungulate prey, livestock and free-ranging dogs. We

considered only those signs that could be unambiguously identified to avoid issues with false positive detections [25], and kept the survey duration short enough to maintain uniform detection conditions [26]. Data from sign surveys were used for modelling carnivore distributions.

Combining interview data with occupancy analyses is now commonplace for examining species distributions over large areas [14,27,28]. We conducted questionnaire-based interview surveys of local residents at the same spatial scale as the sign surveys, from September 2015 to January 2016. Villages and households were selected so as to ensure maximum spatial coverage within each site, and respondents were shown four photographs of each focal species for identification. Conditional on correct identification, surveyors recorded information regarding (i) species presence in or near their village, and (ii) depredation incident in their household or village, attributed to one of the five species. Considering the recall period and accuracy of respondents in such surveys [29], we only considered events pertaining to the previous 12 months. The number of people interviewed per site ranged from 1 to 9, and each interview was treated as an independent spatial replicate. In cases where respondents could not identify the species, the interview was discarded and the replicate was considered as 'non-surveyed' for the purpose of this analysis. We additionally recorded information on socio-economic status of the respondents, land and livestock holdings, economic losses related to depredation, mitigation measures employed and tolerance/acceptance of wildlife. Data from interview surveys were used for modelling depredation patterns.

## 2.5. Analytical methods

### 2.5.1. Modelling carnivore distributions

We fitted detection/non-detection data from sign surveys to single species occupancy models [30]. As the surveys were conducted along contiguous trail segments, we used an extension of the original model described in [31], which accounts for potential spatial correlation of sign detections. We estimate two key parameters from these surveys: $\psi$—probability of species presence in a site, and $p$—probability of detecting a species in a site, given presence. These two parameters were modelled as functions of species-specific ecological covariates identified in the study framework (table 1). In some cases (e.g. dhole), data were too sparse for fitting the correlated-detections model, so we relied on the traditional single species model that assumes independence of replicates ([30]; electronic supplementary material, table S1).

### 2.5.2. Multi-state models to estimate depredation probability

We assessed patterns of depredation by the carnivores applying multi-state occupancy models to data from interview surveys [32]. The detection matrix included '0'—non-detection of a carnivore species by respondent, '1'—detection of carnivore presence but no depredation (state 1), and '2'—detection of carnivore-related depredation event by the respondent (state 2). We estimate the following parameters: $\psi_{P}$—probability of species presence in a site (without depredation); $\psi_{d}$—probability of depredation in a site; $p_{PP}$—probability of detecting species presence in a site; $p_{dd}$—probability of detecting depredation in a site; $p_{pd}$—probability of detecting only presence although there may be depredation in the site. To maintain parsimony and avoid issues with overfitting, we retained an intercept-only effect for $\psi_{P}$ and used the number of interviews per site, i.e. survey effort, as a covariate for detection parameters across all models (electronic supplementary material, table S2). The main parameter of interest $\psi_{d}$ was modelled as a function of ecological and social covariates relevant to each carnivore species (table 2).

### 2.5.3. Dog occupancy and wild carnivore richness

Free-ranging dogs, indirectly facilitated by anthropogenic activities, pose threats to wild carnivores through competition and spreading diseases [33]. Interactions between free-ranging dogs and wild carnivores were assessed in three ways. First, we used frequency of dog signs in each site as a covariate in modelling distributions of the wild carnivores (table 1). Second, we estimated occupancy of dogs and compared its overlap with occupancy of each of the five wild carnivores across 128 sites. Occupancy probability for dogs in the landscape was estimated using the same methods as described above for wild carnivores. Livestock abundance, size of human settlements and human population in each site were used as covariates to model dog occupancy and detectability (see electronic supplementary material, table S3 for model combinations and ranks). Third, we examined the overlap

**Table 1.** Description of covariates used to model probabilities of presence ($\psi$) of the five wild carnivores in the Kanha–Pench forest landscape in 2015–2016; *a priori* predictions for the direction of covariate influence and data sources.

| covariate | species | covariate description and *a priori* prediction | source |
|---|---|---|---|
| chital (chtl) and sambar (smbr) abundance | dhole | Chital *Axis axis* and sambar *Rusa unicolor* are the main wild prey species for dhole. Relative abundance of the two ungulates calculated for each site as ratio of number of replicates with chital or sambar signs to total number of surveyed replicates. Predicted influence: Positive. | data collected during field surveys |
| forest cover (fcov) | dhole, jackal | Land-cover vegetation classes collectively considered as 'forests'. Area under forest cover computed for each site. Predicted influence: Positive. | Indian Institute of Remote Sensing, Govt. of India |
| scrublands (scrb) | wolf, jackal, fox, hyena | Open scrublands are ideal habitats for the four carnivores. Area under scrubland cover computed for each site. Predicted influence: Positive. | Indian Institute of Remote Sensing, Govt. of India |
| agriculture (agri) | wolf, jackal, fox, hyena | Cultivated areas support rodents (prey for jackal and fox), and are used for grazing livestock (prey for wolf and hyena). Proportion of agricultural lands calculated for each site. Predicted influence: Positive. | Indian Institute of Remote Sensing, Govt. of India |
| NDVI (ndvi) | wolf, fox, hyena | The three species generally inhabit dry habitats. Value of mean normalized difference vegetation index (NDVI) during dry months generated for each cell. Predicted influence: Negative. | MODIS/TERRA MOD13Q1 Vegetation Indices |
| terrain ruggedness (rugg) | wolf, jackal, fox, hyena | The four species generally prefer flat terrain. Ruggedness for each site calculated as coefficient of variation (CV) of terrain heterogeneity using Digital Elevation Model maps. Predicted influence: Negative. | Indian Institute of Remote Sensing, Govt. of India |
| dog abundance (dogs) | wolf, jackal, fox | Dogs compete with wild canids for food resources outside protected reserves. Frequency of dog signs calculated for each site as ratio of replicates with dog signs to total number of surveyed replicates. Predicted influence: Negative. | data collected during field surveys |
| livestock abundance (lstk) | dhole, wolf, hyena | Areas with high livestock movement indicate disturbed habitats. Frequency of livestock signs calculated for each site as ratio of replicates with livestock signs to total number of surveyed replicates. Predicted influence: Negative. | data collected during field surveys |

**Table 2.** Description of covariates used to model probabilities of depredation ($\psi_d$) by dhole, wolf and fox in the Kanha–Pench forest landscape in 2015–2016; *a priori* predictions for the direction of covariate influence and data sources.

| covariate | species | covariate description and *a priori* prediction | source |
|---|---|---|---|
| forest cover (fcov) | dhole | Dholes are restricted to forest habitats; depredation events would be within forests. Area under forest cover computed for each site. Predicted influence: Positive. | Indian Institute of Remote Sensing, Govt. of India |
| livestock abundance (lstk) | dhole | Frequency of livestock signs calculated for each site as ratio of replicates with livestock signs to total number of surveyed number of replicates. Predicted influence: Negative. | data collected during field surveys |
| terrain ruggedness (rugg) | dhole | Ruggedness for each site calculated as coefficient of variation (CV) of terrain heterogeneity using Digital Elevation Model maps. Grazing activity would be lower in rough terrain habitats. Predicted influence: Negative. | Indian Institute of Remote Sensing, Govt. of India |
| scrublands (scrb) | wolf, fox | Attacks would be more frequent in habitats preferred by the two carnivores. Area under scrubland cover was computed for each site. Predicted influence: Positive. | Indian Institute of Remote Sensing, Govt. of India |
| settlement size (sett) | wolf, fox | Larger settlements deter carnivore presence; fewer depredation cases. Total area under human settlements was calculated for each site. Predicted influence: Negative. | Indian Institute of Remote Sensing, Govt. of India |
| goat-holding (goat) | wolf | Attacks on goats were mostly attributed to wolves. Average number of goats per household was calculated for each site. Predicted influence: Positive. | questionnaire surveys of households |
| poultry-holding (ptry) | fox | Attacks on poultry were mostly attributed to fox and jackal. Average number of poultry per household was calculated for each site. Predicted influence: Positive. | questionnaire surveys of households |
| occupancy probability (occp) | dhole, wolf, fox | Depredation probability could vary based on carnivores' occupancy or use of a site. Occupancy probabilities were estimated for each site. Predicted influence: Positive/Negative. | estimates from distribution analysis in step 1 |

between estimated dog occupancy and an estimate of wild carnivore richness. Wild carnivore richness index for each site was calculated as:

$$R_i = \sum_{j=1}^{5} \psi_{ij}.$$

Here, richness index $R$ for each site $i$ is the sum of estimated occupancy probability $\psi$ values for each species $j$. Analyses pertaining to all three sections described above were performed in program PRESENCE v. 11.9 [34]. Model selection followed standard protocols for parsimony and multi-model inference [35].

# 3. Results

## 3.1. Distribution and habitat use

We invested a total of 1631 km of walk effort and detected wolf, dhole, jackal, fox and hyena in 16, 4, 64, 23 and 9 sites, respectively. A single model did not fully explain the observed patterns for any of the species. We therefore model-averaged across all candidate models to obtain species-specific estimates of $\psi$ and $p$. Parameter estimates for all species are presented in table 3. Probability of use for dhole was estimated at $\psi$ (s.e.) = 0.12 (0.01) and for wolf at $\psi$ (s.e.) = 0.57 (0.02). Occupancy probability was $\psi$ (s.e.) = 0.86 (0.01) for jackal, 0.50 (0.004) for fox and 0.36 (0.02) for hyena. Forest cover and relative abundance of chital were positively associated with dhole presence. Scrublands were important for wolf, jackal and hyena. Terrain ruggedness appeared to influence wolf and jackal presence, and wolf and fox used drier areas. Influence of livestock abundance was negative for dhole but positive for hyena. For all covariates in the analysis, the direction of influence was indicative rather than conclusive because 95% CI of regression co-efficients straddled 0 (table 4). Frequency of dog signs did not show any effect on the occupancy probability of wild carnivores (table 4). Spatial patterns of carnivore distributions are shown in figure 3.

## 3.2. Depredation patterns and determinants

Depredation probability models were based on data from 675 interviews with local residents. Depredation incidents were recorded from 68 sites for wolf, nine sites for dhole and 44 sites for fox. There were no records of depredation by hyena, but incidents attributed to jackal were reported in more than 95% of the sites. These two species were excluded from modelling depredation and we could perform analyses only for wolf, dhole and fox. Estimated depredation probability was highest for wolf and least for dhole (table 4; figure 4). As with the distribution analysis in the previous step, the direction of covariates' influence was indicative but not conclusive because 95% CI of regression co-efficients straddled 0 (table 5). Depredation by dhole was associated with higher forest cover, lower livestock abundance and higher habitat-use probability (estimated in the previous step). Extent of scrublands, settlement size, habitat-use probability and goat-holding by local residents influenced depredation by wolf; settlement size, occupancy probability and poultry-holding by local residents influenced depredation by fox.

## 3.3. Overlap with free-ranging dogs

Dog signs were detected in 68 of 128 sites. Model-averaged estimate of dog occupancy probability was $\psi$ (s.e.) = 0.84 (0.004), and the detection probability was $p$ (s.e.) = 0.32 (0.01); see figure 5. Dog occupancy appeared to be positively influenced by size of human settlements [$\beta$ (s.e.) = 0.74 (0.68)] and detectability was positively influenced by livestock abundance [$\beta$ (s.e.) = 0.37 (0.14)]. Human population size and livestock abundance did not have a significant influence on dog occupancy (electronic supplementary material, table S3). Examining species-specific overlaps, dog occupancy was positively correlated with occupancy of wolf, jackal, fox and hyena (Pearson's correlation $r$ = 0.32, 0.42, 0.53 and 0.63, respectively), and negatively with dhole ($r$ = −0.73; figure 5). Wild carnivore richness index ranged from 1.38 to 3.41 per site, and these estimates showed a positive relationship with dog occupancy ($r$ = 0.45; figure 5).

**Table 3.** Model-based parameter estimates of probability of presence ($\psi$), probability of presence-only without conflict ($\psi_p$), depredation probability ($\psi_d$), and associated detection probabilities (see Material and methods for full parameter definitions). Parameters $\psi$ and $p$ are for data from sign surveys, and relate to the four-month duration from October 2015 to January 2016. Parameters $\psi_p$, $\psi_d$, $p_{pp}$, $p_{dd}$ and $p_{dp}$ are for data from questionnaire-based interview surveys, and relate to a 1 year time period. Values in parentheses are standard error estimates.

| | $\Psi$ | $p$ | $\psi_p$ | $\psi_d$ | $p_{pp}$ | $p_{dd}$ | $p_{dp}$ |
|---|---|---|---|---|---|---|---|
| wolf | 0.57 (0.02) | 0.03 (0.001) | 0.09 (0.01) | 0.84 (0.01) | 0.90 (0.04) | 0.57 (0.0002) | 0.21 (0.0004) |
| dhole | 0.12 (0.01) | 0.31 (0.02) | 0.53 (0.002) | 0.21 (0.01) | 0.35 (0.004) | 0.25 (0.001) | 0.26 (0.001) |
| jackal | 0.86 (0.01) | 0.79 (0.02) | — | — | — | — | — |
| fox | 0.50 (0.004) | 0.92 (0.02) | 0.23 (0.01) | 0.77 (0.01) | 0.36 (0.09) | 0.19 (0.001) | 0.52 (0.001) |
| hyena | 0.36 (0.02) | 0.02 (0.002) | — | — | — | — | — |

**Table 4.** Estimated β-coefficient values (standard errors in parentheses) for ecological covariates influencing probabilities of carnivore presence $\psi$ in the Kanha−Pench landscape in 2015−2016. Values presented indicate the magnitude and direction of the influence of covariates on carnivore presence probability (based on sign surveys). For all five species, estimates are from the model where the corresponding covariate first appears (based on AIC). chtl-chital abundance; smbr—sambar abundance; fcov—forest cover; scrb—scrubland cover; agri—agricultural land; ndvi—normalized difference vegetation index; rugg—terrain ruggedness; dogs—abundance of free-ranging dogs; lstk—abundance of livestock; all covariates were z-transformed prior to analyses.

| | chtl | smbr | fcov | scrb | agri | ndvi | rugg | dogs | lstk |
|---|---|---|---|---|---|---|---|---|---|
| wolf | — | — | — | 2.76 (1.56) | 0.04 (0.54) | −1.27 (0.79) | 2.04 (1.62) | −0.08 (0.76) | −0.18 (0.70) |
| dhole | 0.63 (0.46) | 0.36 (0.42) | 1.17 (0.99) | — | — | — | — | — | −0.87 (0.51) |
| jackal | — | — | −0.47 (0.54) | 4.08 (2.78) | 0.12 (0.51) | — | 1.02 (1.00) | 0.82 (5.67) | — |
| fox | — | — | — | 0.07 (0.44) | 0.27 (0.36) | −0.33 (0.31) | 0.03 (0.51) | −0.05 (0.25) | — |
| hyena | — | — | — | 2.62 (2.57) | 0.76 (1.47) | −0.15 (0.76) | 2.02 (2.39) | — | 2.69 (1.69) |

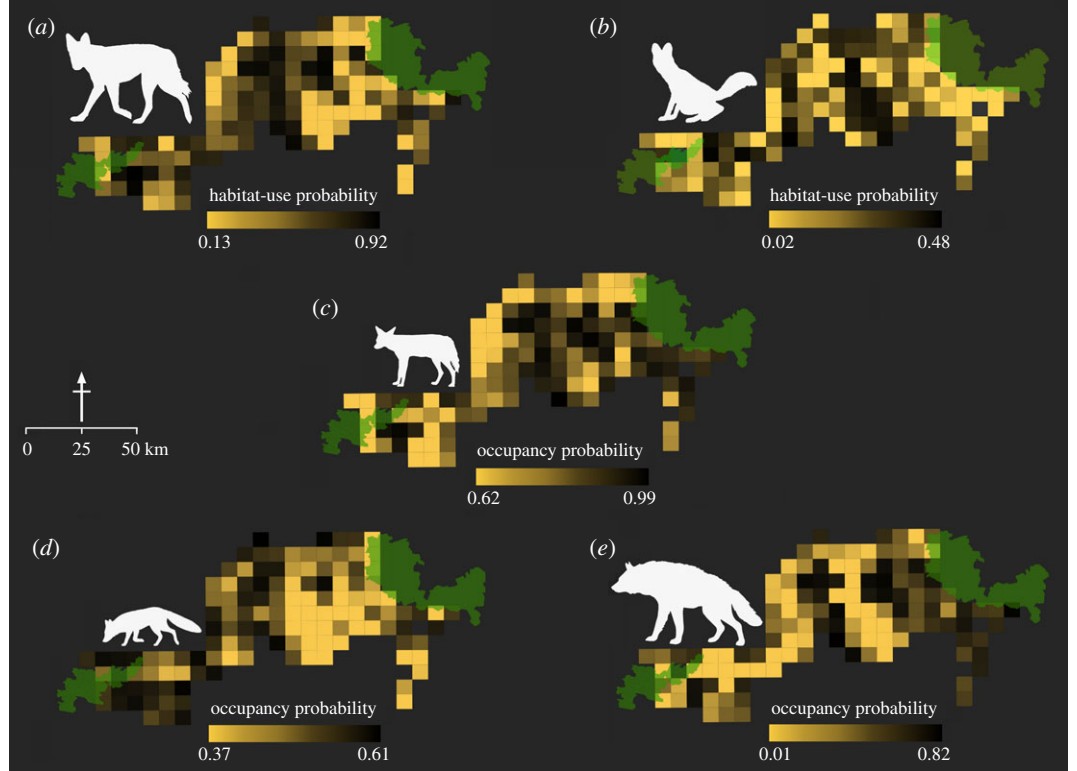

**Figure 3.** Estimated habitat-use probabilities for (*a*) wolf, (*b*) dhole, and true occupancy probabilities for (*c*) jackal, (*d*) fox, and (*e*) hyena, based on sign surveys across 128 sites in the Kanha – Pench forest landscape, 2015 – 2016.

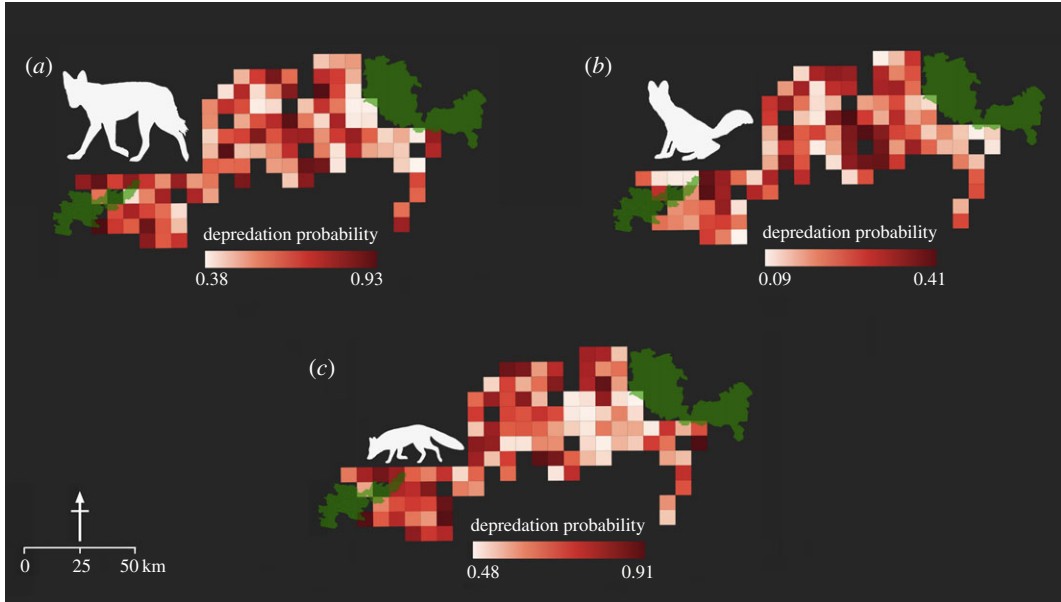

**Figure 4.** Estimated probabilities of depredation by (*a*) wolf, (*b*) dhole, and (*c*) fox, based on interview surveys of local residents across 112 sites in the Kanha – Pench forest landscape, 2015 – 2016. Sites that did not contain villages or households have been clipped out.

## 4. Discussion

The enormous space constraints imposed by a burgeoning human population on wildlife and wild habitats necessitates empirical studies that explore what factors facilitate or deter coexistence between people and carnivores. Conservation biologists and wildlife managers are increasingly recognizing the importance of incorporating human dimensions with ecological knowledge about carnivores to

**Table 5.** Estimated β-coefficient values (standard errors in parentheses) for ecological and social covariates influencing probabilities of livestock/poultry depredation by carnivores ($\psi_d$) in the Kanha−Pench landscape in 2015−2016. Values presented indicate the magnitude and direction of the influence of covariates on depredation probabilities (based on questionnaire surveys). Estimates are from the model where the corresponding covariate first appears (based on AIC). fcov—forest cover; scrb—scrubland cover; rugg—terrain ruggedness; sett—area of human settlements; lstk—abundance of livestock; goat—average goat-holding size; ptry—average poultry-holding size; occp—occupancy probability; all covariates were z-transformed prior to analyses.

| | fcov | scrb | rugg | sett | lstk | goat | ptry | occp |
|---|---|---|---|---|---|---|---|---|
| wolf | — | 2.26 (1.66) | — | −1.06 (0.55) | — | 1.29 (0.87) | — | 0.67 (0.58) |
| dhole | 0.64 (0.48) | — | −0.06 (0.44) | — | −0.57 (0.44) | — | — | 0.86 (0.70) |
| fox | — | −0.28 (0.34) | — | −1.16 (0.81) | — | — | 1.57 (0.88) | 0.68 (0.49) |

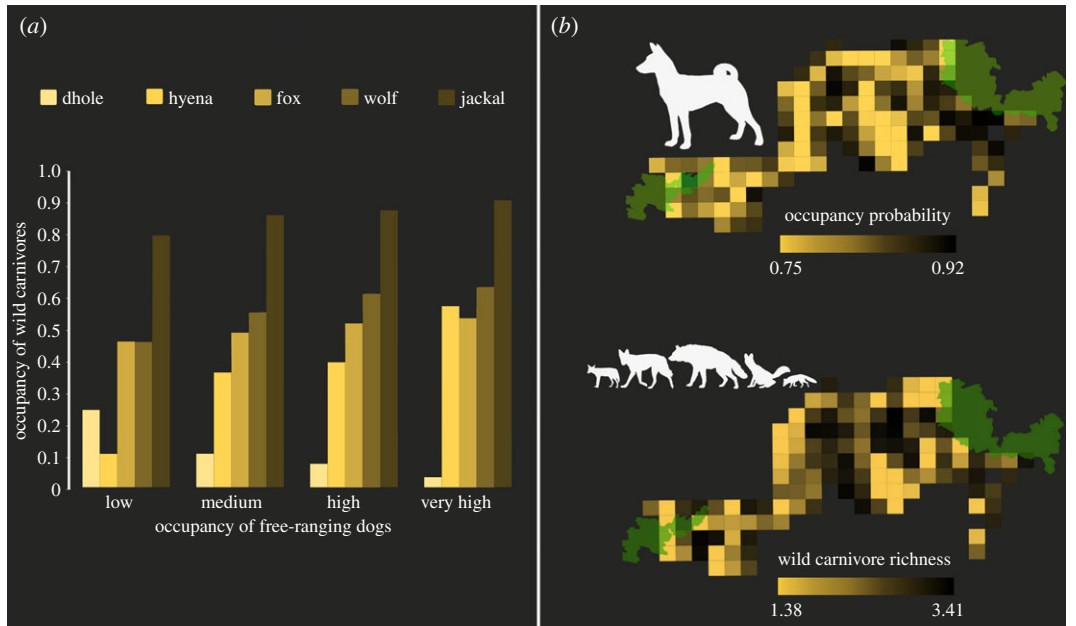

**Figure 5.** (a) Average occupancy of wild carnivores across sites grouped by average dog occupancy values categorized as: low ($\psi = 0.78$), medium ($\psi = 0.82$), high ($\psi = 0.86$), very high ($\psi = 0.90$); (b) comparison between spatial patterns of occupancy of free-ranging dogs and wild carnivore species richness index in the Kanha–Pench forest landscape.

understand these interactions [36,37]. Our study explicitly examined socio-ecological contexts within which a suite of under-studied carnivores co-occur with people at a spatial scale relevant to regional- and national-level policy implications.

## 4.1. Consolidating habitats for carnivores

Dholes typically require intact forests with relatively high wild prey densities and low livestock grazing pressure [19,38]. The Kanha–Pench forest landscape is a potential sink habitat for dholes, linking the two reserves which host source populations [39]. Forests constitute more than 80% of the habitats in our study area, yet dholes used only 12% of the sites. Concomitant impacts of livestock grazing [40,41], fragmentation of forests [42] and recent infrastructural developments in the region [43] could severely paralyse connectivity for dholes in the landscape. Scrublands were important for wolf, jackal and hyena presence. Preserving scrublands entails a host of complexities unlike the case with forests. This is reflected in the gross under-representation of grasslands and scrublands in terrestrial protected reserves across the world [44]. In India, the problem is twofold: (i) scrublands are treated as unproductive 'wastelands', converted into agricultural areas or diverted for commercial use [45], and (ii) most scrublands are revenue lands that are not managed by the forest department. Even when such lands are brought under the department's jurisdiction, they are subverted for highly unscientific and often counter-productive afforestation initiatives. Agricultural lands may serve as supplementary habitats for wolf, jackal, fox and hyena, but our results in this aspect were inconclusive (but see [14,20,23,24]). We submit that our estimates do not reflect the true contribution of agricultural lands as habitats, and this could be an artefact of the spatial scale and location of our sampling units.

## 4.2. Depredation risk in shared landscapes

Livestock depredation is a key factor that can deter human–carnivore coexistence [46]. Mitigating livestock depredation by carnivores, and evaluation of pre-emptive or redressal actions towards losses have received much academic and conservation focus [47,48]. Understanding the spatial risk factors determining depredation incidents is an important tool in carnivore management and conservation. Our approach (i) relied on records of depredation events self-reported by people, (ii) treated 'depredation' as a probabilistic state of carnivore presence (*sensu* [15]), (iii) accounted for potential biases of unequal survey effort or under-reporting/non-detection of depredation records, and (iv)

allowed for explaining observed patterns through a combination of ecological and social attributes of the system. Based on this, we were able to ascertain a probability of depredation risk for each site.

We classify dhole and hyena as 'low conflict-risk' species in our study landscape. Dholes avoided areas with high livestock movement, and probability of dhole presence-only (without depredation) was the highest among three species ($\psi_p = 53\%$). Depredations by dholes are likely from incidental attacks, when cattle are grazed inside forests. Similar trends of low human–dhole conflict have been reported from the Western Ghats [49], but not in Northeast India [50] or Southeast Asia [51]; a potential explanation is that dholes do not attack livestock if there is adequate availability of wild prey. We did not record any reports of depredation by hyenas. This may corroborate observations that hyenas are scavengers and do not actively hunt domestic prey [52]. Based on our results, we classify wolf, jackal and fox as 'high conflict-risk' species. Wolves mostly preyed on goats, and depredation was higher in sites with large scrubland areas and small settlements. As predicted, goat depredation by wolf was also higher in sites with bigger goat-holdings (number of owned goats in our study area ranged from 1 to 30 per household). Jackals generally attacked poultry; jackal-related events were reported by respondents in more than 95% of the sites, precluding us from conducting formal analysis to estimate depredation probability. Poultry depredation by foxes, on the other hand, was skewed towards sites with smaller settlements and high poultry-holdings. Our results are similar to those reported by Karanth *et al.* [53], who found that wolf and jackal were responsible for most depredation-related livestock losses in a part of the same landscape.

## 4.3. Mitigation strategies and compensatory solutions

The efficacy of mitigation measures to avoid depredation, and the financial investments towards these efforts have been a subject of considerable debate. In a recent review, Eklund *et al.* [54] show that most interventions for mitigating human–carnivore conflicts globally have been futile, and argue for evidence-based measures to reduce losses. Within our study area, people used rudimentary fencing structures, moved animals into secured sheds during night time, kept watch dogs, or, maintained fires at night to prevent depredation of livestock. In contrast, poultry were generally free-ranging and kept indoors/in baskets at night. Frequent depredation events also coerced some people into completely giving up livestock ownership. The Government of India mandates compensation administered through the State forest departments for depredation-related losses, with compensation amount varying across States and based on the type of livestock [55]. In the State of Madhya Pradesh (our study location), for example, the current policy compensates loss of cattle (approx. 250–460 USD depending on age and quality) and goats (approx. 46 USD), but not poultry. Respondents generally refrained from claiming compensation for goats because it is nearly impossible to establish proof of depredation by wild canids. Furthermore, a few cases of crop loss were also associated with canids (e.g. maize and tuber consumption by jackal and fox), but compensation for such losses does not feature in the current policy.

State-provisioned monetary compensation is among the most widely used mitigation measures for livestock losses [56]. Monetary compensation initiatives have the potential to alter people's tolerance and acceptance of carnivores [57]. However, flawed implementation makes it an ineffective strategy, particularly in the case of wild canids. Although improvement in husbandry practices—more secure shelters and better herding practices—could reduce depredation rates [58], we believe that decentralized, village-level insurance schemes need to be explored as alternative strategies (e.g. [59]). We note, however, that respondents in our study were generally indifferent towards losses to wild canids, often viewing depredation incidents as inevitable occupational hazards. Such non-negative attitudes may also be attributed to the fact that wild canids in this region do not attack, injure or kill humans (in contrast to tigers or leopards). Nonetheless, we do recognize that we lacked the requisite expertise for making qualitative, in-depth evaluations of human attitudes and perceptions.

## 4.4. Latent threats from free-ranging dogs

Free-ranging dogs are currently the most widespread large carnivore species in the world [60]. With populations persisting largely due to anthropogenic resource-provisioning, dogs can bear a host of negative impacts on wildlife, domestic livestock and human health. Besides their competitive dominance, they also act as reservoirs of lethal diseases, which pose serious threats to wildlife populations [61]. Studies examining dog–wildlife interactions have rightly cautioned about the range of associated risks, substantiating the global efforts invested towards vaccination and sterilization of dogs [33]. The urban, semi-rural and rural landscapes of India support large populations of

free-ranging dogs [62]. In our study, dog occupancy was correlated with larger settlements, perhaps because of high resource provisioning in the form of garbage dumps or voluntary feeding in such sites [63]; although the covariates we use did not receive adequate statistical support. We suspect this is because of the ubiquitous presence of dogs across the landscape. Dogs showed high overlap with overall wild carnivore richness index. Examining these overlaps with individual species, only dholes tended to avoid areas with high dog occupancy (figure 5). Of particular interest is the high overlap between dogs and the other four species. Negative interactions between dogs and wild carnivores typically manifest through interference competition at fine spatial scales [64]. Although there is high overlap at the scale of our sample units, the wild carnivores stand to lose out on many microhabitats within regions where they co-occur with dogs, effectively reducing the total available habitat for wild carnivores. With nearly no predator-imposed control, and very few factors deterring their survival in human-dominated landscapes, dog populations are likely to grow exponentially in the future. Controlling populations of unowned dogs in the country has thus far been unsuccessful, and is often at crossroads with opposing ideologies meted out by advocates of animal rights and welfare groups [65]. We foresee this as a serious concern for survival and persistence of carnivore populations (and their wild prey species) in our study region and similar landscapes across the country.

## 4.5. Implications for conservation

The current focus and investment towards large-scale infrastructure development in the Kanha–Pench forest landscape renders carnivore conservation in a 'triage' scenario [66]. There is also considerable ease with which public lands—including forests—are currently being diverted for commercial use and infrastructure development. Based on our findings, we propose that (i) efforts be directed towards retaining the current land cover structure, configuration and heterogeneity to conserve this carnivore community at the regional scale, as well as planning infrastructure projects and developmental activities so as to facilitate persistence of the five species; (ii) our predictions of spatial depredation patterns be used for conservation prioritization [67] and systematically identifying locations for investment of funds for conflict resolution; and (iii) active control and management of free-ranging dog populations would benefit both humans and wild carnivores. Carnivores provide several direct and indirect benefits to human health and well-being [68]. Considering the high proportion of rodents in the diet of two mesocarnivores in our study (jackal and fox), they may also provide economic benefits through pest control (e.g. [69]). But formulating national policy frameworks for conserving predators whose global ranges are large but nonetheless face local extinctions can be a challenge. As a consequence, governments currently do not recognize the ecological implications of conserving these carnivores. Other than the dhole, all other species in our study are categorized under Near Threatened or Least Concern in the IUCN Red List. But the consortium of habitats they represent are much more fragile than intact forest reserves. We therefore argue for a shift in perspective from the current single-species/wilderness focus to a multi-pronged approach that balances human well-being while also conserving a community of carnivores in shared landscapes [2,70]. Our approach of combining social and ecological dimensions therefore provides insights on how governments and wildlife biologists can adopt alternative strategies for making assessments and planning carnivore conservation, beyond the current protected reserve-based framework.

Ethics. Madhya Pradesh State Forest Department provided necessary research permits to carry out the study. Sign surveys were completely non-invasive and did not involve capture or handling of animals; animal care and use committee approval was not required. Interviews were conducted following verbal consent of local residents.
Data accessibility. Data available from the Dryad Digital Repository: https://doi.org/10.5061/dryad.q3t310k [71].
Authors' contributions. M.P. and A.S. conceived the ideas; A.S., M.P., K.K.K. and N.S.K. designed the survey methodology; M.P. and I.P. collected the data; A.S. and M.P. analysed the data; A.S. and M.P. led the writing of the manuscript. All authors contributed critically to the drafts and gave final approval for publication.
Competing interests. We have no competing interests.
Funding. The DeFries-Bajpai Foundation, Rufford Foundation, Ravi Sankaran Inlaks Fellowship and IDEA-Wild funded the study. During analyses and drafting of the manuscript, A.S. was supported by Wildlife Conservation Society's Christensen Conservation Leaders Scholarship and Wildlife Conservation Network's Sidney Byers Fellowship; A.S. and M.P. were supported by the University of Florida; K.K.K. was supported by Oracle.
Acknowledgements. We are grateful to Madhya Pradesh State Forest Department for providing research permits and supporting this study. We thank J.D. Nichols and J.E. Hines for advice and analytical support. Centre for Wildlife Studies and Wildlife Conservation Society-India programme provided institutional and logistical support. We acknowledge S. Sharma and K. Yadav for assistance in data processing and preliminary analysis. We thank

R. Shukla, M. Agarwala, M. Pariwakam, R. Parameshwaran and K.U. Karanth for their inputs. We are grateful to M. Kumar, S. Hegde, R. Singh, S. Patro, A. Sharma, H. Patel, A.S. Chauhan, M. Babu, P. James, A. Sivaraman, H. Dahodwala, N. Bhatt, S. Gupta, C. Bhatt, A. Raina, N. Salian, V. Patel, H. Singh, P. Sneha, E. Sharma, A. Vaidyanathan, N. Abdulla, A. Agrawal, Shubham, J. Kalaskar, Dinesh, V.T. Ravi, S. Gupta, A. Menon, T. Menon, V. Rawat, S. Tanwar, D. Bhatt, A. Patil, P. Chaudhary, R. Singh, K. Trivedi, and S. Sahu who assisted in data collection.

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
