## [Reviewer comments · Royal Society Open Science]

Review History

RSOS-181041.R0 (Original submission)

Review form: Reviewer 1

Is the manuscript scientifically sound in its present form?

No

Are the interpretations and conclusions justified by the results?

No

Is the language acceptable?

Yes

Is it clear how to access all supporting data?

Yes

Do you have any ethical concerns with this paper?

No

Have you any concerns about statistical analyses in this paper?

Yes

Recommendation?

Reject

Comments to the Author(s)

This is a well written manuscript on an important topic for global carnivore conservation. The authors use an interesting approach incorporating both wildlife ecology and human dimensions to understand habitat use by canids, and relate that back to human-wildlife conflicts, before making inferences on habitat conservation/protected area expansion in a working landscape in India. However, I have major concerns about the results of this study that undermine the findings. For the wild canid occupancy modeling effort, the 95% confidence intervals around the beta estimates for all of the covariates for each species include 0. Moreover, the standard errors are larger than the beta estimates in many cases. This is a clear indication that the results are poor, at best, and any inferences based on these results may be spurious. Furthermore, the authors indicate on line 225 in the text that covariate-specific estimates of detection and occupancy probabilities are model averaged estimates. However, the caption for table 4 indicates that the estimates are from the model in which a covariate first appears. Model averaged estimates would be most appropriate to report, given the model selection uncertainty that is present in most cases.

The estimates of site-level richness are calculated from the species-specific occupancy modeling effort, and these results are not acceptable from a

Based on the model-selection tables for feral dog occupancy, the constant model has at least equal explanatory power as the next best model explaining occupancy for this species, which is human settlement. The authors do not address this in the text, and any estimates of predicted feral dog occupancy should be model averaged estimates. It is not clear in the text if model averaged estimates are reported, and the 95% confidence interval for the reported beta estimate for the influence of human settlements on feral dog occupancy includes 0.

Based on the results of the Depredation Patterns and Determinants occupancy modeling effort, this is the strongest attribute of the manuscript. If the editors decide to invite revisions to the manuscript, the revised manuscript should be based solely on this modeling effort.

Review form: Reviewer 2**Is the manuscript scientifically sound in its present form?**

Yes

Are the interpretations and conclusions justified by the results?

Yes

Is the language acceptable?

Yes

Is it clear how to access all supporting data?

Yes

Do you have any ethical concerns with this paper?

No

Have you any concerns about statistical analyses in this paper?

I do not feel qualified to assess the statistics

Recommendation?

Accept with minor revision (please list in comments)

Comments to the Author(s)

Please see attached file (Appendix A).

Review form: Reviewer 3 (Josip Kusak)

Is the manuscript scientifically sound in its present form?

Yes

Are the interpretations and conclusions justified by the results?

Yes

Is the language acceptable?

Yes

Is it clear how to access all supporting data?

Yes

Do you have any ethical concerns with this paper?

No

Have you any concerns about statistical analyses in this paper?

I do not feel qualified to assess the statistics

Recommendation?

Accept with minor revision (please list in comments)

Comments to the Author(s)

I have found a need for some minor clarifications of methodology (see Appendix B).

The work is otherwise straightforward, clear and understandable, with interesting integration of ecological and socio-economical approach which may be used elsewhere for the improvement of conservation and management of wild canids.

Review form: Reviewer 4

Is the manuscript scientifically sound in its present form?

Yes

Are the interpretations and conclusions justified by the results?

Yes

Is the language acceptable?

Yes

Is it clear how to access all supporting data?

Yes

Do you have any ethical concerns with this paper?

No

Have you any concerns about statistical analyses in this paper?

No

Recommendation?

Major revision is needed (please make suggestions in comments)

Comments to the Author(s)

Journal: Royal Society Open Science

Manuscript ID: RSOS-181041

Manuscript title: A socio-ecological framework for examining human-carnivore interactions: Sympatric wild canids in India as case study

This manuscript examines the relationship between carnivore distribution and patterns of livestock depredation in multi-use heterogeneous landscapes. The authors build upon an occupancy modeling framework combined with structured interviews towards village residents, to assess the impacts of carnivore depredation on husbandry activities and how environmental and social factors help explain interaction patterns and to map conflict-prone areas. The manuscript is well written, for which I applaud the authors, the language is clear, and sections are well structured. Sampling design is robust and occupancy models seem to have been properly fitted. Both occupancy and detection parameters seem to have been carefully thought. However, dealing with some issues must preclude publication, such as the translation of results into insights to land-use policy, strategies for conflict mitigation or conservation of shared landscapes (step 4 of the study framework). Some details on methods need to be clarified, to allow reproducibility, even if the complete dataset is available (as is the case in this manuscript). Overall, this is a good manuscript that I would like to see published, once the concerns I present are resolved.

Specific issues:

Title: The title suggests that the authors are originally describing a new method, in a methodological paper on the use of this innovative socio-ecological framework, which is not the case. As far as I understood, the authors build upon an existing method, to evaluate a biological question with conservation-related purposes.

Abstract:

P2 L46-47: Which results? Which conclusions?

Introduction:

P4 L93: "We examined factors that facilitate coexistence..." This phrase should be re-written, because, as it is, it indicates factors that prevent conflict. If this is not what you mean by it, please re-write.

P4 L100: The use of ", and," with two commas should be avoided throughout the text.

Materials and methods:

L110: The use of semicolon in this phrase should be checked.

Figure 1

Step 1: Please check the arrow that combines Ecological requirements Integrated multi-state occupancy models: if the ecological/environmental predictors have already been used during the

single-species occupancy modeling phase, then the outcome of such models (the carnivore distribution) should not be included in another model (multi-state occupancy models) in addition to the ecological/environmental predictors once again. This double incorporation will result in overfitted models, due to correlation (spatial or environmental) among predictors, and spurious relationships among environmental/ecological predictors and carnivore occupancy. If this is not the case (environmental predictors have not been included twice), please check the text and clarify this section.

Step 3: If carnivore distribution is an input of integrated multi-state occupancy models, shouldn't it be presented in box 2?

Field survey section

I had to read this section a couple of times to understand that single-season and multi-state occupancy models had not been calibrated with the same data (scat signs or interviews). Please clarify this section, explicitly indicating which data belong to which modelling framework. In addition, I suggest the elaboration of another figure, in a conceptual or workflow scheme, to allow better understanding of methods and procedures, especially because there are several and different modelling steps, which use and require different data inputs.

Dog occupancy and wild carnivore richness

Model selection procedures were not explained. How did authors evaluate and select among top-ranking models? A table of species-specific models, showing covariates combinations, could be incorporated, maybe as a supplementary material.

Results

The section describing habitat use/occupancy by wild carnivores is a bit confuse, in my opinion. There are many relationships, so maybe re-writing and organizing the results per species could improve readability.

P8 - L238: Hyenas are notorious by their necrophagous behavior, so I wonder why the authors included it as a target species in a paper on the factors that contribute to human-carnivore conflict... If the scavenger behavior of hyenas is notorious and unambiguous, was there some indication, maybe personal experience in the field observing hyenas preying on livestock or poultry, that would justify the inclusion of such species? If there was not, I believe that the a priori expectation of a relationship is not reasonable and that the species should be removed.

P9 L253-257: The lack of relationship between carnivore richness and dog occupancy, in addition to the species avoidance/congruence in space, is an interesting result, that was overlooked in the Discussion section.

Discussion

P9 L268-272: The authors argue that dhole occupancy was low, even though "forests constitute more than 80% of the habitats" in the study area, but I wonder what the occupancy of dholes on intact forests is. If dholes are generally rare, even on preserved habitat, there is no reason to expect its high occupancy on any landscape.

The section on the interaction between domestic dogs and wild carnivores should be elaborated, especially related to the fact that species do not avoid each other in the scale proposed.

P12 L365-368: The discussion on separation into micro-habitats is speculative, because the authors do not present evidence (results or literature) that overlap at landscape scale may result in habitat differentiation in more local scales.

P10 L300-305: Nonetheless, dhole occupancy was overall low, and hyenas are not expected to prey on animal husbandry (or are they?).

P13 L378: Please explain the "triage" scenario.

Overall, maybe it would be better to avoid excessive adjectives, such as ardent (L381), formidable (L391).

Decision letter (RSOS-181041.R0)

04-Oct-2018

Dear Mr Srivathsa:

Manuscript ID RSOS-181041 entitled "A socio-ecological framework for examining human-carnivore interactions: Sympatric wild canids in India as a case study" which you submitted to Royal Society Open Science, has been reviewed. The comments from reviewers are included at the bottom of this letter.

In view of the criticisms of the reviewers, the manuscript has been rejected in its current form. However, a new manuscript may be submitted which takes into consideration these comments.

Please note that resubmitting your manuscript does not guarantee eventual acceptance, and that your resubmission will be subject to peer review before a decision is made.

Your resubmitted manuscript should be submitted by 03-Apr-2019. If you are unable to submit by this date please contact the Editorial Office.

Please note that Royal Society Open Science will introduce article processing charges for all new submissions received from 1 January 2018. Charges will also apply to papers transferred to Royal Society Open Science from other Royal Society Publishing journals, as well as papers submitted as part of our collaboration with the Royal Society of Chemistry (<http://rsos.royalsocietypublishing.org/chemistry>). If your manuscript is submitted and accepted for publication after 1 Jan 2018, you will be asked to pay the article processing charge, unless you request a waiver and this is approved by Royal Society Publishing. You can find out more about the charges at <http://rsos.royalsocietypublishing.org/page/charges>. Should you have any queries, please contact openscience@royalsociety.org.

on behalf of Dr Michael Tobler (Associate Editor) and Prof. Kevin Padian (Subject Editor)
openscience@royalsociety.org

Associate Editor Comments to Author (Dr Michael Tobler):
Associate Editor: 1
Comments to the Author:

We have received the feedback from four reviewers. All reviewers agreed that this is an interesting study, but one reviewer points out some substantive problems, questioning the robustness of the models and the inferences than can be drawn from them. If the authors can address the concerns and suggestion from the reviewers, I would welcome a resubmission of the manuscript to RSOS.

Reviewers' Comments to Author:

Reviewer: 1

Comments to the Author(s)

This is a well written manuscript on an important topic for global carnivore conservation. The authors use an interesting approach incorporating both wildlife ecology and human dimensions to understand habitat use by canids, and relate that back to human-wildlife conflicts, before making inferences on habitat conservation/protected area expansion in a working landscape in India. However, I have major concerns about the results of this study that undermine the findings. For the wild canid occupancy modeling effort, the 95% confidence intervals around the beta estimates for all of the covariates for each species include 0. Moreover, the standard errors are larger than the beta estimates in many cases. This is a clear indication that the results are poor, at best, and any inferences based on these results may be spurious. Furthermore, the authors indicate on line 225 in the text that covariate-specific estimates of detection and occupancy probabilities are model averaged estimates. However, the caption for table 4 indicates that the estimates are from the model in which a covariate first appears. Model averaged estimates would be most appropriate to report, given the model selection uncertainty that is present in most cases.

The estimates of site-level richness are calculated from the species-specific occupancy modeling effort, and these results are not acceptable from a

Based on the model-selection tables for feral dog occupancy, the constant model has at least equal explanatory power as the next best model explaining occupancy for this species, which is human settlement. The authors do not address this in the text, and any estimates of predicted feral dog occupancy should be model averaged estimates. It is not clear in the text if model averaged estimates are reported, and the 95% confidence interval for the reported beta estimate for the influence of human settlements on feral dog occupancy includes 0.

Based on the results of the Depredation Patterns and Determinants occupancy modeling effort, this is the strongest attribute of the manuscript. If the editors decide to invite revisions to the manuscript, the revised manuscript should be based solely on this modeling effort.

Reviewer: 2

Comments to the Author(s)

Please see attached file.

Reviewer: 3

Comments to the Author(s)

I have found a need for some minor clarifications of methodology (attached PDF)

The work is otherwise straightforward, clear and understandable, with interesting integration of ecological and socio-economical approach which may be used elsewhere for the improvement of conservation and management of wild canids.

Reviewer: 4

Comments to the Author(s)

Journal: Royal Society Open Science

Manuscript ID: RSOS-181041

Manuscript title: A socio-ecological framework for examining human-carnivore interactions: Sympatric wild canids in India as case study

This manuscript examines the relationship between carnivore distribution and patterns of livestock depredation in multi-use heterogeneous landscapes. The authors build upon an occupancy modeling framework combined with structured interviews towards village residents, to assess the impacts of carnivore depredation on husbandry activities and how environmental and social factors help explain interaction patterns and to map conflict-prone areas. The manuscript is well written, for which I applaud the authors, the language is clear, and sections are well structured. Sampling design is robust and occupancy models seem to have been properly fitted. Both occupancy and detection parameters seem to have been carefully thought. However, dealing with some issues must preclude publication, such as the translation of results into insights to land-use policy, strategies for conflict mitigation or conservation of shared landscapes (step 4 of the study framework). Some details on methods need to be clarified, to allow reproducibility, even if the complete dataset is available (as is the case in this manuscript). Overall, this is a good manuscript that I would like to see published, once the concerns I present are resolved.

Specific issues:

Title: The title suggests that the authors are originally describing a new method, in a methodological paper on the use of this innovative socio-ecological framework, which is not the case. As far as I understood, the authors build upon an existing method, to evaluate a biological question with conservation-related purposes.

Abstract:

P2 L46-47: Which results? Which conclusions?

Introduction:

P4 L93: "We examined factors that facilitate coexistence..." This phrase should be re-written, because, as it is, it indicates factors that prevent conflict. If this is not what you mean by it, please re-write.

P4 L100: The use of ", and," with two commas should be avoided throughout the text.

Materials and methods:

L110: The use of semicolon in this phrase should be checked.

Figure 1

Step 1: Please check the arrow that combines Ecological requirements? Integrated multi-state occupancy models: if the ecological/environmental predictors have already been used during the single-species occupancy modeling phase, then the outcome of such models (the carnivore distribution) should not be included in another model (multi-state occupancy models) in addition to the ecological/environmental predictors once again. This double incorporation will result in overfitted models, due to correlation (spatial or environmental) among predictors, and spurious relationships among environmental/ecological predictors and carnivore occupancy. If this is not the case (environmental predictors have not been included twice), please check the text and clarify this section.

Step 3: If carnivore distribution is an input of integrated multi-state occupancy models, shouldn't it be presented in box 2?

Field survey section

I had to read this section a couple of times to understand that single-season and multi-state occupancy models had not been calibrated with the same data (scat signs or interviews). Please clarify this section, explicitly indicating which data belong to which modelling framework. In addition, I suggest the elaboration of another figure, in a conceptual or workflow scheme, to

allow better understanding of methods and procedures, especially because there are several and different modelling steps, which use and require different data inputs.

Dog occupancy and wild carnivore richness

Model selection procedures were not explained. How did authors evaluate and select among top-ranking models? A table of species-specific models, showing covariates combinations, could be incorporated, maybe as a supplementary material.

Results

The section describing habitat use/occupancy by wild carnivores is a bit confuse, in my opinion. There are many relationships, so maybe re-writing and organizing the results per species could improve readability.

P8 - L238: Hyenas are notorious by their necrophagous behavior, so I wonder why the authors included it as a target species in a paper on the factors that contribute to human-carnivore conflict... If the scavenger behavior of hyenas is notorious and unambiguous, was there some indication, maybe personal experience in the field observing hyenas predated livestock or poultry, that would justify the inclusion of such species? If there was not, I believe that the a priori expectation of a relationship is not reasonable and that the species should be removed.

P9 L253-257: The lack of relationship between carnivore richness and dog occupancy, in addition to the species avoidance/congruence in space, is an interesting result, that was overlooked in the Discussion section.

Discussion

P9 L268-272: The authors argue that dhole occupancy was low, even though "forests constitute more than 80% of the habitats" in the study area, but I wonder what the occupancy of dholes on intact forests is. If dholes are generally rare, even on preserved habitat, there is no reason to expect its high occupancy on any landscape.

The section on the interaction between domestic dogs and wild carnivores should be elaborated, especially related to the fact that species do not avoid each other in the scale proposed.

P12 L365-368: The discussion on separation into micro-habitats is speculative, because the authors do not present evidence (results or literature) that overlap at landscape scale may result in habitat differentiation in more local scales.

P10 L300-305: Nonetheless, dhole occupancy was overall low, and hyenas are not expected to prey on animal husbandry (or are they?).

P13 L378: Please explain the "triage" scenario.

Overall, maybe it would be better to avoid excessive adjectives, such as ardent (L381), formidable (L391).

Author's Response to Decision Letter for (RSOS-181041.R0)

See Appendix C.

RSOS-182008.R0

Review form: Reviewer 1

Is the manuscript scientifically sound in its present form?

No

Are the interpretations and conclusions justified by the results?

No

Is the language acceptable?

Yes

Is it clear how to access all supporting data?

Yes

Do you have any ethical concerns with this paper?

No

Have you any concerns about statistical analyses in this paper?

Yes

Recommendation?

Major revision is needed (please make suggestions in comments)

Comments to the Author(s)

The authors have disregarded my major concerns with their interpretation of results, therefore, I still recommend major revisions or rejection. In their response, the authors state that studies of large carnivore occupancy rarely produce statistically significant results, particularly when covariates are Z-transformed prior to analysis. This is simply not the case. Moreover, Z-transformation of covariates usually increases the performance of Occupancy models, and lowers SE values.

As for model averaging of beta estimates for individual covariates, see Grueber et al. 2011, *Journal of Evolutionary Biology*.

While I agree that betas with an SE including 0 can indicate the direction of a relationship, there is so much uncertainty in the estimates presented in tables 4 & 5 that even the direction of the relationship is in question. On the extreme end, how can the direction of a relationship be determined with a confidence interval of -1.02 - 1.1 (wolf/agri; table 4) be interpreted? In their comments, the authors agree that there is very little support for a covariate when the SE is larger than the beta estimate for that covariate. Why are those covariates still included in the tables, then? At a bare minimum, please remove those estimates from the tables. I.E., sambar, agriculture, and dogs, should not be included as a covariate for any of the models in Table 4. The model for wolf should not include livestock, the model for jackal should not include forest cover, the model for fox should not include scrub or ruggedness, and the model for hyena should only include livestock. The same standards should be applied to the models presented in table 5. New predictions will need to be made from models not containing the covariates that need to be removed, and the results and discussion should be modified to reflect all of the above changes.

Review form: Reviewer 4

Is the manuscript scientifically sound in its present form?

Yes

Are the interpretations and conclusions justified by the results?

Yes

Is the language acceptable?

Yes

Is it clear how to access all supporting data?

Yes

Do you have any ethical concerns with this paper?

No

Have you any concerns about statistical analyses in this paper?

No

Recommendation?

Accept as is

Comments to the Author(s)

The authors have made the necessary changes and the paper should be accepted for publication on its current form.

Decision letter (RSOS-182008.R0)

31-Jan-2019

Dear Mr Srivathsa,

The Subject Editor assigned to your paper ("A socio-ecological framework for examining human-carnivore interactions: Sympatric wild canids in India as a case study") has now received comments from reviewers. We would like you to revise your paper in accordance with the referee and Associate Editor suggestions which can be found below (not including confidential reports to the Editor). Please note this decision does not guarantee eventual acceptance.

Please submit a copy of your revised paper before 23-Feb-2019. Please note that the revision deadline will expire at 00.00am on this date. If we do not hear from you within this time then it will be assumed that the paper has been withdrawn. In exceptional circumstances, extensions may be possible if agreed with the Editorial Office in advance. We do not allow multiple rounds of revision so we urge you to make every effort to fully address all of the comments at this stage. If deemed necessary by the Editors, your manuscript will be sent back to one or more of the original reviewers for assessment. If the original reviewers are not available we may invite new reviewers.

When submitting your revised manuscript, you must respond to the comments made by the referees and upload a file "Response to Referees" in "Section 6 - File Upload". Please use this to document how you have responded to each of the comments, and the adjustments you have made. In order to expedite the processing of the revised manuscript, please be as specific as possible in your response.

- Ethics statement

- Data accessibility

If you wish to submit your supporting data or code to Dryad (<http://datadryad.org/>), or modify your current submission to dryad, please use the following link:
<http://datadryad.org/submit?journalID=RSOS&manu=RSOS-182008>

- Competing interests

- Authors' contributions

- Acknowledgements

- Funding statement

on behalf of Dr Michael Tobler (Associate Editor) and Professor Kevin Padian (Subject Editor)
openscience@royalsociety.org

Reviewer comments to Author:

Reviewer: 1

Comments to the Author(s)

The authors have disregarded my major concerns with their interpretation of results, therefore, I still recommend major revisions or rejection. In their response, the authors state that studies of large carnivore occupancy rarely produce statistically significant results, particularly when covariates are Z-transformed prior to analysis. This is simply not the case. Moreover, Z-transformation of covariates usually increases the performance of Occupancy models, and lowers SE values.

As for model averaging of beta estimates for individual covariates, see Grueber et al. 2011, *Journal of Evolutionary Biology*.

While I agree that betas with an SE including 0 can indicate the direction of a relationship, there is so much uncertainty in the estimates presented in tables 4 & 5 that even the direction of the relationship is in question. On the extreme end, how can the direction of a relationship be determined with a confidence interval of -1.02 - 1.1 (wolf/agri; table 4) be interpreted? In their comments, the authors agree that there is very little support for a covariate when the SE is larger than the beta estimate for that covariate. Why are those covariates still included in the tables, then? At a bare minimum, please remove those estimates from the tables. I.E., sambar, agriculture, and dogs, should not be included as a covariate for any of the models in Table 4. The model for wolf should not include livestock, the model for jackal should not include forest cover, the model for fox should not include scrub or ruggedness, and the model for hyena should only include livestock. The same standards should be applied to the models presented in table 5. New predictions will need to be made from models not containing the covariates that need to be removed, and the results and discussion should be modified to reflect all of the above changes.

Reviewer: 4

Comments to the Author(s)

The authors have made the necessary changes and the paper should be accepted for publication on its current form.

Author's Response to Decision Letter for (RSOS-182008.R0)

See Appendix D.

RSOS-182008.R1 (Revision)

Review form: Reviewer 4

Is the manuscript scientifically sound in its present form?

Yes

Are the interpretations and conclusions justified by the results?

Yes

Is the language acceptable?

Yes

Is it clear how to access all supporting data?

Yes

Do you have any ethical concerns with this paper?

No

Have you any concerns about statistical analyses in this paper?

No

Recommendation?

Accept as is

Comments to the Author(s)

Authors have made necessary changes and so I recommend the acceptance of the manuscript on its current form.

Decision letter (RSOS-182008.R1)

07-May-2019

Dear Mr Srivathsa,

I am pleased to inform you that your manuscript entitled "Examining human-carnivore interactions using a socio-ecological framework: Sympatric wild canids in India as a case study" is now accepted for publication in Royal Society Open Science.

Royal Society Open Science operates under a continuous publication model (<http://bit.ly/cpFAQ>). Your article will be published straight into the next open issue and this will be the final version of the paper. As such, it can be cited immediately by other researchers.

As the issue version of your paper will be the only version to be published I would advise you to check your proofs thoroughly as changes cannot be made once the paper is published.

on behalf of Dr Michael Tobler (Associate Editor) and Kevin Padian (Subject Editor)
openscience@royalsociety.org

Associate Editor Comments to Author (Dr Michael Tobler):

The revised manuscript has been evaluated by one of the previous reviewers. Based on the feedback, I recommend that this manuscript is accepted for publication.

Reviewer comments to Author:
Reviewer: 4

Comments to the Author(s)
Authors have made necessary changes and so I recommend the acceptance of the manuscript on its current form.

Appendix A

Royal Society Open Science RSOS-181041: A socio-ecological framework for examining humancarnivore interactions: Sympatric wild canids in India as a case study

This paper employs a relatively novel approach, combining ecological and social data to identify factors that facilitate and impede coexistence between humans and four wild canid species in India, and in turn, provide refreshingly practical suggestions for canid conservation. I think that this is a high-quality piece of research and my comments are very minor. I am however unqualified to critique the statistical approach used and recommend that a reviewer with experience in this type of analysis be consulted.

Minor revisions

L41: It would be useful to mention the survey method used. Also, it is unclear here whether you mean that occupancy of both dholes and jackals was influenced by habitat type, or that the variation in occupancy between dholes and jackals is due to habitat type (rather than interspecific differences in occupancy).

L47: Some explanation of what you mean by 'spatial prioritisation', similar to that at L384, would be useful here.

L98: It may be useful here to summarise in a few words what the approach of Galvez et al. involves (for readers unfamiliar with this area).

L107: suggest inserting 'canid' before 'species'.

L124: lat/long would be useful.

L156: Is there any evidence that the focal species use (or avoid) roads/tracks? Direct sightings were excluded to maintain uniformity in detection but if species differentially use roads/tracks this issue would still arise.

L158: I'm impressed that enough scat and track signs could be unambiguously assigned to species. I was going to suggest that a description of these signs and how you went about this would be useful but I agree, the reference to Karanth et al. is adequate. It may however be useful to indicate the body sizes of your study species and how similar/different their signs are. You could perhaps indicate what proportion of records were discarded due to ambiguity in ID.

Appendix B**ROYAL SOCIETY
OPEN SCIENCE****A socio-ecological framework for examining human-carnivore interactions: Sympatric wild canids in India as a case study**

Journal:	Royal Society Open Science
Manuscript ID	RSOS-181041
Article Type:	Research
Date Submitted by the Author:	29-Jun-2018
Complete List of Authors:	Srivathsa, Arjun; Centre for Wildlife Studies, ; Wildlife Conservation Society, India Program; University of Florida, School of Natural Resources and Environment; University of Florida, Department of Wildlife Ecology and Conservation Puri, Mahi; Wildlife Conservation Society, India Program; Centre for Wildlife Studies, Karanth, Krithi; Columbia University, Ecology, Evolution and Environmental Biology Patel, Imran; Centre for Wildlife Studies Kumar, Narayanrao; Centre for Wildlife Studies, ; Wildlife Conservation Society - India program,
Subject:	ecology < BIOLOGY
Keywords:	carnivores, coexistence, depredation, interviews, occupancy modelling, sign surveys
Subject Category:	Biology (whole organism)

**Title: A socio-ecological framework for examining human-carnivore interactions:**
**Sympatric wild canids in India as a case study**

**Running head: People and predators in shared habitats**

**Authors:** Arjun Srivathsa^{1,2,3,4}, Mahi Puri^{2,3,4}, Krithi K. Karanth^{4,5,6}, Imran Patel⁴, N. Samba
Kumar³

**Author Affiliations:**

1 School of Natural Resources and Environment, University of Florida, Gainesville, FL, USA

2 Department of Wildlife Ecology and Conservation, University of Florida, Gainesville, FL,
USA

3 Wildlife Conservation Society, India Program, Bengaluru, India

4 Centre for Wildlife Studies, Bengaluru, India

5 Wildlife Conservation Society, New York, NY, USA

6 Environmental Science and Policy, Nicholas School of the Environment, Duke University,
Durham, NC, USA

**Corresponding author:** Arjun Srivathsa (arjuns.cws@gmail.com)

Abstract

Many carnivore species inhabit human-dominated landscapes outside protected reserves.
Spatially explicit assessments of carnivore distributions and livestock depredation patterns in
human-use landscapes are crucial for minimising negative interactions and fostering coexistence
between people and predators. India harbors 23% of the world's carnivore species that share
space with 1.3 billion people in ~2.3% of the global land area. We examined carnivore
distributions and human-carnivore interactions in a multi-use forest landscape in central India,
focusing on five sympatric carnivore species: Indian gray wolf *Canis lupus pallipes*, dhole *Cuon*
*alpinus*, Indian jackal *C. aureus indicus*, Indian fox *Vulpes bengalensis* and striped hyena
*Hyaena hyaena*. Carnivore occupancy varied from 12% for dholes to 86% for jackals, influenced
by forests, open scrublands, and terrain ruggedness. Livestock/poultry depredation probability in
the landscape ranged from 21% for dholes to >95% for jackals, influenced by land cover and
livestock- or poultry-holding. The five species also showed high spatial overlap with free-
ranging dogs, suggesting potential competitive interactions and disease-risks, with consequences
for human health and safety. Our results provide insights on factors that facilitate and impede co-
occurrence between people and predators. Spatial prioritisation of carnivore-rich areas and
conflict-prone locations could facilitate human-carnivore coexistence in shared habitats. The
framework we use is ideally suited for making socio-ecological assessments of human-carnivore
interactions in other multi-use landscapes and regions, worldwide.

[revised manuscript text omitted]

Estimates ranged from probability of use for dhole at ψ (SE) = 0.12 (0.01) to probability of
occupancy for jackals at ψ (SE) = 0.86 (0.01). Forest cover and principal prey abundance were
positively associated with dhole presence. Scrublands were important for wolf, jackal, and
hyena. Terrain ruggedness influenced wolf and jackal presence, and, wolf and fox preferred drier
areas. Influence of livestock abundance was negative for dhole but positive for hyena. Frequency
of dog signs did not have a significant effect on the occupancy probability of wild carnivores
(table 4). Spatial patterns of carnivore distributions are shown in figure 3.

3.2. Depredation patterns and determinants

Depredation probability models were based on data from 675 interviews with local residents.
Depredation incidents were recorded from 68 sites for wolf, 9 sites for dhole, and 44 sites for
fox. There were no records of depredation by hyena, but incidents attributed to jackal were
reported in >95% of the sites. We could perform formal analyses only for wolf, dhole and fox.
Estimated depredation probability was highest for wolf and least for dhole (table 4; figure 4).
Depredation by dhole was associated with higher forest cover, lower livestock abundance, and
higher habitat-use probability (estimated in the previous step). Extent of scrublands, settlement
size, habitat-use probability, and goat-holding by local residents influenced depredation by wolf;

settlement size, occupancy probability and poultry-holding by local residents influenced
depredation by fox (table 5).

**3.3. Overlap with free-ranging dogs**

Dog signs were detected in 68 of 128 sites. We estimated dog occupancy probability at ψ (SE) =
0.84 (0.004), and detection probability at p (SE) = 0.32 (0.01); see figure 5. Dog occupancy was

[revised manuscript text omitted]

**Supplementary material**

**Tables S1, S2, and S3** show comparisons of models used for estimating carnivore distribution,
depredation, and occupancy of free-ranging dogs, respectively.

**Data accessibility**

The data used in this manuscript are available at Dryad:

<https://datadryad.org/review?doi=doi:10.5061/dryad.hq7b13d>

**Competing interests**

We have no competing interests.

**Author contributions**

415 A.S. and M.P. conceived the ideas; A.S., M.P., K.K.K. and N.S.K designed the survey
methodology; M.P. and I.P. collected the data; A.S. and M.P. analysed the data; A.S. and M.P.
led the writing of the manuscript. All authors contributed critically to the drafts and gave final
approval for publication.

**Ethics statement**

Madhya Pradesh State Forest Department provided necessary research permits to carry out the
study. Sign surveys were completely non-invasive and did not involve capture or handling of
animals; animal care and use committee approval was not required. Interviews were conducted
following verbal consent of local residents.

**Acknowledgements**

We are grateful to Madhya Pradesh State Forest Department for providing research permits and
supporting this study. We thank J.D. Nichols and J.E. Hines for advice and analytical support.

Centre for Wildlife Studies and Wildlife Conservation Society-India program provided

institutional and logistical support. We acknowledge S. Sharma and K. Yadav for assistance in
data processing and preliminary analysis. We thank R. Shukla, M. Agarwala, M. Pariwakam, R.
Parameshwaran and K.U. Karanth for their inputs. We are grateful to M. Kumar, S. Hegde, R.
Singh, S. Patro, A. Sharma, H. Patel, A. S. Chauhan, M. Babu, P. James, A. Sivaraman, H.
Dahodwala, N. Bhatt, S. Gupta, C. Bhatt, A. Raina, N. Salian, V. Patel, H. Singh, P. Sneha, E.
Sharma, A. Vaidyanathan, N. Abdulla, A. Agrawal, Shubham, J. Kalaskar, Dinesh, V.T. Ravi, S.
Gupta, A. Menon, T. Menon, V. Rawat, S. Tanwar, D. Bhatt, A. Patil, P. Chaudhary, R. Singh,
437 K. Trivedi, and S. Sahu who assisted in data collection.

**Funding**

The DeFries-Bajpai Foundation, Rufford Foundation, Ravi Sankaran Inlaks Fellowship and
IDEA-Wild funded the study. During analyses and drafting of the manuscript, A.S. was
supported by Wildlife Conservation Society's Christensen Conservation Leaders Scholarship and
Wildlife Conservation Network's Sidney Byers Fellowship; A.S. and M.P. were supported by the
University of Florida; K.K.K was supported by Oracle.

**References**

1. Ripple WJ, Estes JA, Beschta RL, Wilmers CC, Ritchie EG, Hebblewhite M, Berger J,
Elmhagen B, Letnic M, Nelson MP, Schmitz OJ. 2014 Status and ecological effects of the
world's largest carnivores. *Science* **343**, 1241484.
2. Carter NH, Linnell JD. 2016 Co-adaptation is key to coexisting with large carnivores. *Trends*
*Ecol Evol* **31**, 575–578.
3. Joppa LN, Pfaff A. 2009 High and far: biases in the location of protected areas. *PLoS One* **4**,
pe8273.
4. Jenkins CN, Joppa LN. 2009 Expansion of the global terrestrial protected area system. *Biol.*
*Conserv.* **142**, 2166–2174.
5. Venter O, Fuller RA, Segan DB, Carwardine J, Brooks T, Butchart SH, Di Marco M, Iwamura
457 T, Joseph L, O'Grady D, Possingham HP. 2014 Targeting global protected area expansion
for imperiled biodiversity. *PLoS Biol.* **12**, p.e1001891.
6. Chapron G, López-Bao JV. 2014 Conserving carnivores: politics in play. *Science* **343**, 1199–
1200.

[revised manuscript text omitted]

Poultry-holding (ptry)	fox	Attacks on poultry were mostly attributed to fox and jackal. Average number of poultry per household was calculated for each site. Predicted influence: Positive	Questionnaire surveys of households
Occupancy probability (occp)	dhole, wolf, fox	Depredation probability could vary based on carnivores' occupancy or use of a site. Occupancy probabilities were estimated for each site. Predicted influence: Positive/Negative	Estimates from distribution analysis in step 1

Table 3. Model-based parameter estimates of probability of presence (ψ), probability of presence-only without conflict (ψ_p), depredation probability (ψ_d), and associated detection probabilities (see Materials and Methods for full parameter definitions). Parameters ψ and p are for data from sign surveys, and relate to the four-month duration from October 2015 to January 2016. Parameters ψ_p , ψ_d , p_{pp} , p_{dd} , and p_{dp} are for data from questionnaire-based interview surveys, and relate to a one-year time period. Values in parentheses are standard error estimates.

[revised manuscript text omitted]

426x243mm (300 x 300 DPI)

C

Response to reviewers' comments

Associate Editor Comments to Author (Dr Michael Tobler):

Associate Editor: 1

Comments to the Author:

We have received the feedback from four reviewers. All reviewers agreed that this is an interesting study, but one reviewer points out some substantive problems, questioning the robustness of the models and the inferences than can be drawn from them. If the authors can address the concerns and suggestion from the reviewers, I would welcome a resubmission of the manuscript to RSOS.

Response: Thank you very much. We really appreciate the constructive and positive comments provided by the four reviewers. We have addressed each comment in this document and revised the manuscript accordingly. We have also provided justifications or clarifications as relevant. Our responses are in boldface font and the reviewer comments are in italicized text.

Reviewers' Comments to Author:

Reviewer: 1

Comments to the Author(s)

This is a well written manuscript on an important topic for global carnivore conservation. The authors use an interesting approach incorporating both wildlife ecology and human dimensions to understand habitat use by canids, and relate that back to human-wildlife conflicts, before making inferences on habitat conservation/protected area expansion in a working landscape in India.

Response: Thank you very much.

However, I have major concerns about the results of this study that undermine the findings. For the wild canid occupancy modeling effort, the 95% confidence intervals around the beta estimates for all of the covariates for each species include 0. Moreover, the standard errors are larger than the beta estimates in many cases. This is a clear indication that the results are poor, at best, and any inferences based on these results may be spurious.

Response: We agree. The 95% CI for all covariates do straddle 0. We would like to assert, however, that in large mammal occupancy assessments across large landscapes (particularly when covariates are z-transformed) very few studies are able to show statistically conclusive influence of predictor variables (i.e., 95% CI of beta-coefficients do not overlap 0). Like most other studies of similar nature, our results are *indicative* of the direction of effect. We do acknowledge that in cases where SE is greater than the mean, there is very little support for the covariate. Our inferences therefore do not apply to any covariate where the SE is larger than the mean for the corresponding regression coefficient.

Furthermore, the authors indicate on line 225 in the text that covariate-specific estimates of detection and occupancy probabilities are model averaged estimates. However, the caption for table 4 indicates that the estimates are from the model in which a covariate first appears. Model averaged estimates would be most appropriate to report, given the model selection uncertainty that is present in most cases.

Response: We follow model-averaging methods as described in Burnham and Anderson (2002) to obtain final estimates of the key parameters (Ψ and Ψ_d). Given the range of issues with model-averaging individual regression coefficients (sensu Cade 2015; Banner and Higgs

2017; Dormann et al. 2018), we have refrained from doing so. The caption for Table 4 is correct in its current form. No change made.

The estimates of site-level richness are calculated from the species-specific occupancy modeling effort, and these results are not acceptable from a

Response: This comment was not complete so we are not fully sure of what the reviewer intends to convey. Nevertheless, we agree that the richness value is not based on the most ideal method, although the approach we use has been applied in other studies (e.g., see Karanth 2011 *Current Science*; Calabrese et al. 2014 *Global Ecology and Biogeography*; Einoder et al. 2018 *Plos One*). There are several recent studies that propose better methods to estimating species richness using an occupancy framework. We were constrained to using this “index of richness” because of the models we used for separately estimating habitat use and depredation, which are ideally suited for the corresponding objectives. We have revised the text and now refer to ‘richness’ as ‘richness index’.

Based on the model-selection tables for feral dog occupancy, the constant model has at least equal explanatory power as the next best model explaining occupancy for this species, which is human settlement. The authors do not address this in the text, and any estimates of predicted feral dog occupancy should be model averaged estimates. It is not clear in the text if model averaged estimates are reported, and the 95% confidence interval for the reported beta estimate for the influence of human settlements on feral dog occupancy includes 0.

Response: Feral dog occupancy estimate is based on model-averaging the ψ parameter. The widespread presence of feral dogs is likely the reason for weak covariate support. We have added this explanation in the revised manuscript.

Based on the results of the Depredation Patterns and Determinants occupancy modeling effort, this is the strongest attribute of the manuscript. If the editors decide to invite revisions to the manuscript, the revised manuscript should be based solely on this modeling effort.

Response: Thank you for recognizing this aspect. We agree that the strength of our paper lies with the analytical approach used, which we believe is relevant and applicable in similar assessments of human-carnivore interactions in other regions and landscapes.

Reviewer: 2

Comments to the Author(s)

This paper employs a relatively novel approach, combining ecological and social data to identify factors that facilitate and impede coexistence between humans and four wild canid species in India, and in turn, provide refreshingly practical suggestions for canid conservation. I think that this is a high-quality piece of research and my comments are very minor. I am however unqualified to critique the statistical approach used and recommend that a reviewer with experience in this type of analysis be consulted.

Response: Thank you very much. It is very encouraging to read positive comments from reviewers and we are glad you find our paper interesting and valuable for wild canid conservation.

Minor revisions

L41: It would be useful to mention the survey method used. Also, it is unclear here whether you mean that occupancy of both dholes and jackals was influenced by habitat type, or that the

variation in occupancy between dholes and jackals is due to habitat type (rather than interspecific differences in occupancy).

Response: We mention dhole and jackal to represent the range of estimates (low to high), to maintain brevity and adherence to the word limit. We have rephrased this sentence for better clarity.

L47: Some explanation of what you mean by ‘spatial prioritisation’, similar to that at L384, would be useful here.

Response: We have provided an explanation in the Discussion section but retained the Abstract text as is to adhere to journal word limits.

L98: It may be useful here to summarise in a few words what the approach of Galvez et al. involves (for readers unfamiliar with this area).

Response: We have provided a brief explanation of the Galvez et al. approach in the revised manuscript.

L107: suggest inserting ‘canid’ before ‘species’.

Response: We have inserted ‘carnivore’ instead of ‘canid’ because the striped hyena is not a canid.

L124: lat/long would be useful.

Response: We have now included Lat/Long of the study location.

L156: Is there any evidence that the focal species use (or avoid) roads/tracks? Direct sightings were excluded to maintain uniformity in detection but if species differentially use roads/tracks this issue would still arise.

Response: We know that dholes use forest roads/trails based on available literature. The sampling design we use here has not been implemented in studies of the other four species. We relied on field knowledge and logistical feasibility to optimize detections of these four species.

L158: I’m impressed that enough scat and track signs could be unambiguously assigned to species. I was going to suggest that a description of these signs and how you went about this would be useful but I agree, the reference to Karanth et al. is adequate. It may however be useful to indicate the body sizes of your study species and how similar/different their signs are. You could perhaps indicate what proportion of records were discarded due to ambiguity in ID.

Response: We have included body sizes (weights) of the focal species in the revised manuscript. None of the records were discarded post-hoc. When signs could not be unambiguously attributed to a particular species, we refrained from recording this information in the field. So, we do not have a number/proportion of records that were discarded. We only recorded detections that could be reliably (to the best of our ability) assigned to a particular species.

Reviewer: 3

Comments to the Author(s)

I have found a need for some minor clarifications of methodology (attached PDF)

The work is otherwise straightforward, clear and understandable, with interesting integration of

ecological and socio-economical approach which may be used elsewhere for the improvement of conservation and management of wild canids.

Response: Thank you very much. We have revised the manuscript incorporating all changes as suggested in the annotated PDF file.

Reviewer: 4

Comments to the Author(s)

Journal: Royal Society Open Science

Manuscript ID: RSOS-181041

Manuscript title: A socio-ecological framework for examining human-carnivore interactions: Sympatric wild canids in India as case study

This manuscript examines the relationship between carnivore distribution and patterns of livestock depredation in multi-use heterogeneous landscapes. The authors build upon an occupancy modeling framework combined with structured interviews towards village residents, to assess the impacts of carnivore depredation on husbandry activities and how environmental and social factors help explain interaction patterns and to map conflict-prone areas. The manuscript is well written, for which I applaud the authors, the language is clear, and sections are well structured. Sampling design is robust and occupancy models seem to have been properly fitted. Both occupancy and detection parameters seem to have been carefully thought. However, dealing with some issues must preclude publication, such as the translation of results into insights to land-use policy, strategies for conflict mitigation or conservation of shared landscapes (step 4 of the study framework). Some details on methods need to be clarified, to allow reproducibility, even if the complete dataset is available (as is the case in this manuscript). Overall, this is a good manuscript that I would like to see published, once the concerns I present are resolved.

Response: Thank you very much.

Specific issues:

Title: The title suggests that the authors are originally describing a new method, in a methodological paper on the use of this innovative socio-ecological framework, which is not the case. As far as I understood, the authors build upon an existing method, to evaluate a biological question with conservation-related purposes.

Response: Thank you for pointing this out. We have restructured the title such that it better reflects the main theme of our study.

Abstract:

P2 L46-47: Which results? Which conclusions?

Response: We have rephrased this sentence.

Introduction:

P4 L93: “We examined factors that facilitate coexistence...” This phrase should be re-written, because, as it is, it indicates factors the prevent conflict. If this is not what you mean by it, please re-write.

Response: Here, our aim was to convey that although there are certain factors that determine patterns of depredation, the focal species do coexist alongside people. This pertains to habitats or land-cover types that are conducive to carnivores inhabiting human-dominated areas. Massive land-use changes or land conversions will be detrimental to persistence of these carnivores. The current landscape matrix and associated ecological attributes are thus important factors facilitating human-carnivore coexistence. We deliberate on these aspects in the Discussion section.

P4 L100: The use of “, and,” with two commas should be avoided throughout the text.

Materials and methods:

Response: Revised as suggested, throughout the manuscript.

L110: The use of semicolon in this phrase should be checked.

Response: We have removed the semicolon and revised these sentences.

Figure 1

Step 1: Please check the arrow that combines Ecological requirements \diamond Integrated multi-state occupancy models: if the ecological/environmental predictors have already been used during the single-species occupancy modeling phase, then the outcome of such models (the carnivore distribution) should not be included in another model (multi-state occupancy models) in addition to the ecological/environmental predictors once again. This double incorporation will result in overfitted models, due to correlation (spatial or environmental) among predictors, and spurious relationships among environmental/ecological predictors and carnivore occupancy. If this is not the case (environmental predictors have not been included twice), please check the text and clarify this section.

Response: There are some covariates that were predicted to influence distribution/habitat-use but also likely to influence depredation patterns, and therefore used in both steps. We checked for cross-correlations before running the models and refrained from using highly correlated predictors ($r > /0.7$) in the same model (to avoid issues of over-fitting models and multi-collinearity among predictor variables).

Step 3: If carnivore distribution is an input of integrated multi-state occupancy models, shouldn't it be presented in box 2?

Response: Step 3 includes results from model predictions. Carnivore distributions are also stand-alone results and are included in Step 3. We have used these estimates as potential predictors in modelling depredation patterns, and thus an arrow feeds back from Step 3 to Step 2.

Field survey section

I had to read this section a couple of times to understand that single-season and multi-state occupancy models had not been calibrated with the same data (scat signs or interviews). Please clarify this section, explicitly indicating which data belong to which modelling framework. In addition, I suggest the elaboration of another figure, in a conceptual or workflow scheme, to allow better understanding of methods and procedures, especially because there are several and different modelling steps, which use and require different data inputs.

Response: The sign-based surveys were conducted to examine distribution patterns and the interview surveys for depredation patterns. These are described in two separate paragraphs. We have retained these as two paragraphs and edited the text to make it more explicit in the revised manuscript.

Dog occupancy and wild carnivore richness

Model selection procedures were not explained. How did authors evaluate and select among top-ranking models? A table of species-specific models, showing covariates combinations, could be incorporated, maybe as a supplementary material.

Response: Model selection procedure is included in the Results subsection 3.3. Species-specific model comparisons showing covariate combinations are in the Supplementary material (Tables S1, S2 and S3).

Results

The section describing habitat use/occupancy by wild carnivores is a bit confuse, in my opinion. There are many relationships, so maybe re-writing and organizing the results per species could improve readability.

Response: We chose to present results in the current format for brevity and to avoid redundancy (which would be the case if we presented the results species-wise). For better clarity, we have now included key parameter estimates for all species in the Results section.

P8 - L238: Hyenas are notorious by their necrophagous behavior, so I wonder why the authors included it as a target species in a paper on the factors that contribute to human-carnivore conflict... If the scavenger behavior of hyenas is notorious and unambiguous, was there some indication, maybe personal experience in the field observing hyenas preying on livestock or poultry, that would justify the inclusion of such species? If there was not, I believe that the a priori expectation of a relationship is not reasonable and that the species should be removed.

Response: Information on hyena-human interactions in India is severely inadequate in published literature. Therefore, we included hyena as one of the focal species in this study. None of the survey respondents reported depredation by hyenas, confirming speculations made in previous studies, that hyenas are, in fact, largely scavengers and do not directly prey on livestock.

P9 L253-257: The lack of relationship between carnivore richness and dog occupancy, in addition to the species avoidance/congruence in space, is an interesting result, that was overlooked in the Discussion section.

Response: We have alluded to this result in the Discussion sub-section 4.4. “Latent threats from free-ranging dogs”, and discussed species-specific overlaps thereafter.

Discussion

P9 L268-272: The authors argue that dhole occupancy was low, even though “forests constitute more than 80% of the habitats” in the study area, but I wonder what the occupancy of dholes on intact forests is. If dholes are generally rare, even on preserved habitat, there is no reason to expect its high occupancy on any landscape.

Response: Habitat-use probability by dholes in forest habitats is not necessarily low. For example, Srivathsa et al. (2014) reported an occupancy estimate of 0.71 for dholes in Bandipur National Park, southern India. Based on anecdotal information and field knowledge, we know that there are dhole populations in Kanha and Pench reserves. We expected that our study area— which links the two reserves— would be used by dholes to move between the reserves. Our results indicate that dholes used these areas with very low probability.

P12 L365-368: The discussion on separation into micro-habitats is speculative, because the authors do not present evidence (results or literature) that overlap at landscape scale may result in habitat differentiation in more local scales.

Response: We agree that this is a speculation based on current knowledge of dog-wildlife interactions. Most of the papers cited in this paragraph provide evidence in favour of our claim.

P10 L300-305: Nonetheless, dhole occupancy was overall low, and hyenas are not expected to prey on animal husbandry (or are they?).

Response: These two species share space with humans but are responsible for very little/no damage to livestock. We have accordingly classified them as “low conflict-risk” species.

P13 L378: Please explain the “triage” scenario.

Response: The triage here refers to the previous line and links to the rest of the paragraph. The government of India’s current agenda of rapid infrastructure development puts non-protected wildlife-rich habitats at risk. The triage scenario arises when promoting necessary economic development while also trying to ensure conservation of wildlife and wild habitats in these areas.

Overall, maybe it would be better to avoid excessive adjectives, such as ardent (L381), formidable (L391).

Response: We have removed these words in the revised manuscript.

Comments to the Author(s) based on annotated PDF files:

1. PDF file “journal.pone.0174259.s004.PDF”

L323: Authors introduce here data on livestock husbandry, which was not presented/analyzed in results.

Response: We have modelled livestock-holding by household as a covariate while modelling depredation patterns. Please refer to Table 1. If the reviewer is referring specifically to husbandry practices, kindly note that analysing mitigation measures adopted by local communities and evaluating the efficacy of these measures was beyond the scope of this study.

L333: Explain why it is impossible and what? Is it impossible to distinguish depredation of different canid species?

Response: Depredation by carnivores like tigers and leopards are conspicuous. The track marks, drag marks etc. may be identified easily. In many instances, the livestock carcass is not fully consumed, and bite marks on the carcass can be used for ascertaining carnivore identity. All these aspects are examined by the Forest Department during field visits before issuing compensation. Wild canid signs cannot be easily identified if personnel are not trained to do so (which is usually the case). Wild canids rarely leave carcasses behind nor do they cache them and return to consume the same. These factors make it difficult to retrospectively establish proof of livestock attack and attribute it to specific canid species.

2. PDF file “RSOS_for review_proof_DR”

L253: Using a Pearson’s correlation here doesn’t make any sense. The authors modeled wild carnivore occupancy as a function of dog occupancy, and any relationships between the two should be predicted from those models.

Response: We used encounter frequency of dog signs to model carnivore occupancy. The effect was weak and inconclusive. The Pearson’s correlation here is between dog occupancy (estimated separately) and carnivore occupancy.

L336: Feral dogs are responsible for a great deal of depredations in the US, where their numbers are much lower than those reported in this study. How do we know that losses attributed to

jackals and wolves are not actually caused by dogs, and then potentially scavenged by the wild canids?

Response: We were very cautious while recording information related to depredation by wild canids. Respondents were asked to describe the events in as much detail as possible, and survey teams were trained to recognize/identify species behavior and traits. We readily discounted species attribution by respondents if we thought there was even the slightest ambiguity in species description or details of the depredation event.

L344: This is in stark contrast to stakeholder perceptions of livestock depredation by wild canids in the US. even in areas with no large carnivores that pose a threat to humans.

Response: We agree. There are some studies that delve deeper into human acceptance of large carnivores in shared landscapes of India. As we openly acknowledge in the manuscript, we did not have the expertise to perform detailed analysis of human perceptions and attitudes in this study.

Tables 4 and 5: 95% CI around all beta coefficients includes 0. Moreover, the SE is larger than the coefficient in many cases. Likely don't have enough data for robust inference from these models

Response: We have addressed this in response to a similar comment by Reviewer #1.

Response to reviewer comments

We have split the reviewer's comment into parts and answered each part separately for better clarity.

Reviewer #1:

The authors have disregarded my major concerns with their interpretation of results, therefore, I still recommend major revisions or rejection. In their response, the authors state that studies of large carnivore occupancy rarely produce statistically significant results, particularly when covariates are Z-transformed prior to analysis. This is simply not the case. Moreover, Z-transformation of covariates usually increases the performance of Occupancy models, and lowers SE values..

Response: Thank you very much for your comments. We sincerely apologize that our previous response was not acceptable to you. We have tried our best to explain our stance more clearly in the response below.

As for model averaging of beta estimates for individual covariates, see Grueber et al. 2011, Journal of Evolutionary Biology

Response: We studied the Grueber et al. 2011 paper and its plausible relevance to our study. While their approach to multi-model inference has merit, our overall philosophy aligns more closely with Cade (2015; Ecology), which takes into account the nested nature of covariate models, low to moderate multicollinearity of predictors (as is the case in our study) and the fact that we model-average across all models (rather than only the top models or only across models where a given predictor is present). We respectfully disagree with the reviewer about model-averaging regression co-efficients.

While I agree that betas with an SE including 0 can indicate the direction of a relationship, there is so much uncertainty in the estimates presented in tables 4 & 5 that even the direction of the relationship is in question. On the extreme end, how can the direction of a relationship be determined with a confidence interval of -1.02 - 1.1 (wolf/agri; table 4) be interpreted? In their comments, the authors agree that there is very little support for a covariate when the SE is larger than the beta estimate for that covariate. Why are those covariates still included in the tables, then? At a bare minimum, please remove those estimates from the tables. I.E., sambar, agriculture, and dogs, should not be included as a covariate for any of the models in Table 4. The model for wolf should not include livestock, the model for jackal should not include forest cover, the model for fox should not include scrub or ruggedness, and the model for hyena should only include livestock. The same standards should be applied to the models presented in table 5. New predictions will need to be made from models not containing the covariates that need to be removed, and the results and discussion should be modified to reflect all of the above changes.

Response: For each species, we made a set of *a priori* predictions (based on our knowledge of their ecology and the landscape) and confronted these with data. Some predictors (and models) had better support compared to the others. There was of course a set of predictors for which the effect was completely inconclusive. In Tables 4 and 5 we have presented the beta-estimates for ALL predictors from the model where they first appear, irrespective of whether they received adequate support or not. Removing these would imply that we are only presenting results that corroborate our predictions, and this would be incorrect. We have presented the list of candidate models for all species- for distribution and depredation- in the Supplementary Information. We are sure you will agree that re-running models based on *post hoc* observations of regression results would qualify as dredging.

We reiterate that in the Results section (and subsequently in the Discussion), we have only inferred from and deliberated on those covariates whose SE is lower than the mean of the corresponding predictor variable. But we agree that this was not articulated clearly in the previous version of the MS. We have therefore revised the text in the Results section to explicitly acknowledge the uncertainty around covariate effects.